# Secreted exosomes induce filopodia formation

Caitlin McAtee[1,2†], Mikin R Patel[3†], Daisuke Hoshino[4†], Bong Hwan Sung[1,2], Ariana K von Lersner[1,2,5], Mingjian Shi[3,6], Nan Hyung Hong[1], Anna Young[3‡], Evan Krystofiak[7], Andries Zijlstra[2,5§], Alissa M Weaver[1,2,5]*

[1]Department of Cell and Developmental Biology, Vanderbilt University School of Medicine, Nashville, United States; [2]Center for Extracellular Vesicle Research, Vanderbilt University School of Medicine, Nashville, United States; [3]Department of Biological Sciences, Vanderbilt University, Nashville, United States; [4]Kanagawa Cancer Center, Yokohama, Japan; [5]Department of Pathology, Microbiology and Immunology, Vanderbilt University Medical Center, Nashville, United States; [6]Department of Biomedical Informatics, Vanderbilt University Medical Center, Nashville, United States; [7]Cell Imaging Shared Resource EM Facility, Vanderbilt University, Nashville, United States

*For correspondence:
alissa.weaver@vanderbilt.edu

†These authors contributed equally to this work.

Present address: ‡Baylor College of Medicine, Houston, United States; §StimulusBio, Houston, United States

Competing interest: The authors declare that no competing interests exist.

## eLife Assessment

In this **important** manuscript, the authors reveal novel findings on the role of exosomes in regulating filopodia formation. Filopodia are crucial for various cellular processes, including migration, polarization, directional sensing, and the formation of neuronal synapses. The authors **convincingly** demonstrate that exosomes, particularly those enriched with the protein THSD7A, play a significant role in promoting filopodia formation in both cancer cells and neurons.

**Abstract** Filopodia are dynamic adhesive cytoskeletal structures that are critical for directional sensing, polarization, cell-cell adhesion, and migration of diverse cell types. Filopodia are also critical for neuronal synapse formation. While dynamic rearrangement of the actin cytoskeleton is known to be critical for filopodia biogenesis, little is known about the upstream extracellular signals. Here, we identify secreted exosomes as potent regulators of filopodia formation. Inhibition of exosome secretion inhibited the formation and stabilization of filopodia in both cancer cells and neurons and inhibited subsequent synapse formation by neurons. Rescue experiments with purified small and large extracellular vesicles (EVs) identified exosome-enriched small EVs (SEVs) as having potent filopodia-inducing activity. Proteomic analyses of cancer cell-derived SEVs identified the TGF-β family coreceptor endoglin as a key SEV-enriched cargo that regulates filopodia. Cancer cell endoglin levels also affected filopodia-dependent behaviors, including metastasis of cancer cells in chick embryos and 3D migration in collagen gels. As neurons do not express endoglin, we performed a second proteomics experiment to identify SEV cargoes regulated by endoglin that might promote filopodia in both cell types. We discovered a single SEV cargo that was altered in endoglin-KD cancer SEVs, the transmembrane protein Thrombospondin Type 1 Domain Containing 7A (THSD7A). We further found that both cancer cell and neuronal SEVs carry THSD7A and that add-back of purified THSD7A is sufficient to rescue filopodia defects of both endoglin-KD cancer cells and exosome-inhibited neurons. We also find that THSD7A induces filopodia formation through activation of the Rho GTPase, Cdc42. These findings suggest a new model for filopodia formation, triggered by exosomes carrying THSD7A.

**eLife digest** Cell migration is a crucial process that underpins both healthy functions – such as organ and blood vessel development, and the formation of synaptic connections between neurons – and pathological events like cancer metastasis. For cells to migrate, they must reorganize their cytoskeleton to form specialized structures that support their movement.

This reorganization includes filopodia, which are slender protrusions that play key roles in sensing directional cues at the leading edge of migrating cells and initiating contact with other cells. The extracellular signals that trigger the formation of filopodia remain poorly understood, but recent research suggests that extracellular vesicles may be involved in this process.

Also known as exosomes, these vesicles are membrane-enclosed particles that carry proteins, lipids, and nucleic acids between cells, representing an alternative mode of cellular communication. McAtee et al. aimed to determine whether exosomes could also promote filopodia formation.

They studied two distinct systems: cancer cells, where filopodia facilitate metastasis, and rat neurons, where they promote synapse formation. In both contexts, exosomes emerged as potent, general regulators of filopodia formation, indicating a broad role in controlling this fundamental cellular behavior.

Using genetic inhibition of exosome secretion, combined with add-back of extracellular vesicles, McAtee et al. demonstrated a direct causal relationship between exosomes and filopodia formation. In neurons, filopodia matured into dendritic spines – critical structures for synaptic connectivity between neurons. A quantitative analysis of neurons confirmed that exosomes indeed promoted the formation of dendritic spines and synapses. They also identified the exosomal protein THSD7A as key in driving filopodia formation in both neurons and cancer cells. Moreover, in cancer cells, the secretion of THSD7A-containing exosomes, filopodia formation, cell migration, and metastatic colony formation were regulated by a growth factor receptor known as endoglin.

The study by McAtee et al. highlights a novel role for extracellular vesicles in filopodia formation, enhancing our understanding of how cells migrate in health and disease. Interestingly, both molecules identified as cargoes of filopodia-regulating exosomes are associated with disease: endoglin with hereditary hemorrhagic telangiectasia, and THSD7A with a kidney disease called secondary membranous nephropathy. This suggests that dysregulation of filopodia may occur in these conditions. Furthermore, endoglin and/or THSD7A could serve as potential targets for anti-cancer therapies. Indeed, some promising investigations have already employed anti-endoglin antibodies to inhibit metastasis.

## Introduction

Dynamic rearrangement of the actin cytoskeleton is critical for cell shape changes, elaboration of specialized cell structures, and cell movement. Thin actin-rich structures called filopodia are used for directional sensing and making initial contacts between neighboring cells (*Blake and Gallop, 2023*; *Heckman and Plummer, 2013*; *Mattila and Lappalainen, 2008*). In migrating cells, filopodia protrude from the cell's leading edge to promote cell polarization and facilitate adhesion (*Arjonen et al., 2011*; *Jacquemet et al., 2015*). Filopodia are also important for establishing cell contacts (*Almagro et al., 2010*; *Hammers et al., 2021*; *Heimsath et al., 2017*; *Vasioukhin et al., 2000*). In neurons, filopodia initiate synapse formation by contacting the axon of neighboring neurons and then develop into specialized postsynaptic structures called dendritic spines that are important for learning and memory (*Fiala et al., 1998*; *Kanjhan et al., 2016*; *Wit and Hiesinger, 2023*; *Ziv and Smith, 1996*). Filopodia have also been shown to contribute to invasive cancer cell behaviors, including metastasis (*Arjonen et al., 2011*; *Jacquemet et al., 2015*; *Arjonen et al., 2014*; *Hashimoto et al., 2011*; *Jacquemet et al., 2017*; *Machesky, 2008*; *Machesky and Li, 2010*; *Pan et al., 2023*; *Sun et al., 2011*; *Zhang et al., 2021*).

Several different mechanisms have been described to promote filopodia formation, including direct nucleation of unbranched actin filaments by formins and other nucleators and reorganization of branched cortical actin filaments followed by actin polymerization (*Blake and Gallop, 2023*; *Mattila and Lappalainen, 2008*; *Gallop, 2020*). Additional actin regulatory proteins promote uncapping of the barbed ends, elongation, and bundling of actin filaments. These actin regulatory proteins are

controlled upstream by phosphatidylinositols, especially PI(4,5)P2, PI(3,4,5)P3, and PI(3,4)P2, and Rho GTPases, particularly Cdc42 (*Blake and Gallop, 2023*; *Nobes and Hall, 1995*). Myosin-X may further promote filopodia maintenance by transporting and/or organizing cytoskeletal and adhesion molecules at filopodia tips (*Almagro et al., 2010*; *Bohil et al., 2006*; *Miihkinen et al., 2021*; *Popović et al., 2023*; *Zhang et al., 2004*).

While the intracellular cytoskeletal machinery that drives filopodia formation has been heavily studied, there is little known about the extracellular cues that activate that machinery. Bone morphogenic proteins (BMPs) have been shown to regulate filopodia in endothelial cells, myoblasts, and neurons, often through activation of the Rho GTPase Cdc42 (*Gamell et al., 2008*; *Lee-Hoeflich et al., 2004*; *Wakayama et al., 2015*). VEGFA has also been shown to promote endothelial sprouting and filopodia formation (*Gerhardt et al., 2003*). In neurons, a variety of extracellular molecules have been shown to regulate filopodia formation and dynamics, including brain-derived neurotrophic factor (*Menna et al., 2009*), Slit (*McConnell et al., 2016*), Netrin-1 (*Shekarabi and Kennedy, 2002*), and nerve growth factor (*Ketschek and Gallo, 2010*). Filopodia formation has also been shown to be influenced in tumor cells by TGF-β signaling-induced alterations in gene expression, specifically upregulation of fascin protein (*Sun et al., 2011*). While these studies have identified some extracellular regulators of filopodia formation, the molecular mechanisms are poorly understood, and some are indirect, via inducing gene expression changes in cytoskeletal components.

Exosomes are small extracellular vesicles (EVs) that are formed within multivesicular late endosomes (MVEs) and released into the extracellular space by fusion of MVEs with the plasma membrane. Exosomes carry bioactive cargoes, including proteins, lipids, and nucleic acids and promote autocrine and paracrine cell communication across a variety of systems (*Dixson et al., 2023*; *Maas et al., 2017*; *van Niel et al., 2018*). Notably, exosomes have been shown to promote polarization and motility of multiple cell types, including cancer cells, immune cells, and single-celled amoebae (*Brown et al., 2018*; *Kriebel et al., 2018*; *Majumdar et al., 2021*; *Sung et al., 2015*; *Sung et al., 2021*; *Sung and Weaver, 2017*). Exosomes also play a key role in metastasis by seeding metastatic niches (*Asao et al., 2023*; *Peinado et al., 2012*; *Wortzel et al., 2019*).

In previous studies, we found that exosome secretion is critical for the formation of two motility-related actin cytoskeletal structures: invadopodia and nascent cell-matrix adhesions (*Sung et al., 2015*; *Hoshino et al., 2013*). In our live imaging studies, we also frequently observed filopodia formation in close proximity to sites of exosome secretion and adhesion formation (*Sung et al., 2015*). Here, we directly investigate the role of exosomes in filopodia formation and stability. In both cancer cells and primary rat neurons, we find that genetic inhibition of exosome biogenesis or secretion inhibits filopodia formation and stability. In neurons, the reduction of filopodia was accompanied by a decrease in dendritic spines and synapses, which are structures that develop from filopodia. Filopodial defects of exosome-inhibited cells were fully rescued by the add-back of purified small EVs (SEVs, containing exosomes) but not larger ectosome-type EVs (LEVs). Proteomic comparison of SEVs and LEVs purified from melanoma cells identified the TGF-β co-receptor endoglin as a cargo that is unique to the SEV fraction. Endoglin was found to promote filopodia formation by cancer cells, as well as the filopodia-dependent behaviors of cancer cell metastasis and 3D motility. A quantitative proteomic comparison of exosomes purified from control and endoglin-KD melanoma cells revealed that the filopodia regulatory transmembrane molecule thrombospondin type 1 domain containing 7A (THSD7A) is reduced in endoglin-KD exosomes. Further investigation revealed that THSD7A is present on both cancer cell and neuronal exosomes and that recombinant purified THSD7A is sufficient to rescue filopodia defects resulting from exosome inhibition in both cell types. THSD7A-induced filopodia formation was diminished in the presence of a chemical inhibitor of the small GTPase Cdc42. Altogether, we find that exosomes drive filopodia formation in multiple cell types by carrying the filopodia regulator THSD7A.

Authors' Note: Throughout this paper, we use both the term 'exosomes' and 'small extracellular vesicles (SEVs)'. 'Exosomes' is used when we have perturbed the endocytic pathway to specifically inhibit exosome biogenesis within MVE or secretion (via affecting docking of MVE with the plasma membrane) and can, therefore, attribute the biogenesis of those EVs to the endocytic pathway. 'SEVs' is used for purified EV preparations where we have isolated a heterogeneous population of EVs and cannot define their origin.

## Results

### Exosome markers localize to filopodia

To visualize exosome association with cytoskeletal structures, we stained for CD63 in HT1080 fibrosarcoma cells, which form numerous filopodia. The cells were fixed and co-stained with phalloidin-Alexa Fluor 488 to visualize filamentous actin (F-actin) in filopodia. Confocal images of the cells show localization of CD63 puncta at the tips of filopodia (*Figure 1A*, arrowheads). To observe the dynamic relationship between exosome secretion and filopodia formation, we utilized a live cell reporter that allows us to dynamically visualize the fusion of MVE with the plasma membrane, pHluorin-M153R-CD63-mScarlet (*Sung et al., 2020*). pHluorin is a pH-sensitive form of GFP that is non-fluorescent at acidic pH inside late endosomes and fluoresces at neutral pH (*Miesenböck et al., 1998*), such as occurs upon endosome-plasma membrane fusion, whereas mScarlet is a pH-insensitive red fluorescent protein (*Bindels et al., 2017*) and can report on MVE movements within the cell. Using this probe stably expressed in HT1080 fibrosarcoma cells, we observed localization of MVEs at (red signal) and fusion with (yellow puncta) the plasma membrane immediately before or coincident with filopodia initiation (*Figure 1B*, *Figure 1—video 1*). Quantitative analysis of the timing of yellow fluorescence appearance at the plasma membrane revealed that exosome secretion occurred with a median time of 20 s before filopodia formation (*Figure 1C and D*). We also examined the localization of MVE markers to filopodia in primary cortical neurons by transiently transfecting neurons with the MVE docking factor GFP-Rab27b (*Ostrowski et al., 2010*) along with mCherry as a cytoplasmic 'filler' to visualize cellular neuronal structures. Notably, GFP-Rab27b localized to both the base and tips of neuronal filopodia (*Figure 1E and F*), with more localized to the base than to the tips (63% vs 37%, *Figure 1F*). Together, both the cancer cell and neuron data suggest an association of MVEs and exosomes with filopodia.

### Exosomes promote filopodia formation and stability in cancer cells

To directly test whether exosomes affect filopodia formation and/or stability, shRNA targeting Rab27a or Hrs was expressed in B16F1 melanoma cells (*Figure 2—figure supplement 1A and B*), a frequently used cell line for studying filopodia, and shRNA targeting Rab27a was expressed in HT1080 cells (*Figure 2—figure supplement 2A*). Rab27a is a key factor controlling MVE docking at the plasma membrane, which allows intraluminal vesicles to be secreted as exosomes (*Ostrowski et al., 2010*). The ESCRT-0 protein Hrs plays an important role in the biogenesis of intraluminal vesicles within MVEs (*Dixson et al., 2023*; *Tamai et al., 2010*). The exosome-enriched small EV (SEV) population was isolated by cushion density gradient ultracentrifugation (*Li et al., 2018*), after isolation of large EVs by differential centrifugation from conditioned media (see Materials and methods). Consistent with current standards in the field (*Lötvall et al., 2014*; *Théry et al., 2018*; *Welsh et al., 2024*), the EVs carried typical EV markers and were of the expected size and morphology by nanoparticle tracking analysis (NTA) and negative stain transmission electron microscopy (*Figure 2—figure supplement 1C and E*; *Figure 2—figure supplement 2B and C*). Quantitation by NTA of EVs isolated from conditioned media confirmed that B16F1 Rab27a- and Hrs-knockdown (KD) cells secrete fewer SEVs compared to control cells (*Figure 2A*). Consistent with a selective effect on MVE docking and biogenesis, these constructs had no effect on secretion of LEVs, which should contain primarily shed ectosomes (*Figure 2A*). Phalloidin staining of actin filaments in Rab27a- and Hrs-KD B16F1 cells revealed reduced numbers of filopodia compared with control B16F1 cells (*Figure 2B and C*). Similarly, inhibition of exosome secretion by Rab27a KD in HT1080 fibrosarcoma cells also led to a decrease in filopodia numbers (*Figure 2—figure supplement 2D–F*). Consistent with a specific role for exosomes in controlling filopodia dynamics, the addition of SEVs purified from control B16F1 cells to Hrs-KD B16F1 cells rescued the defect in filopodia numbers (*Figure 2D and E*). By contrast, there was no effect of LEVs on filopodia numbers. The addition of SEVs to B16F1 shScr control cells also increased filopodia number in a dose-dependent manner, while LEVs had no effect (*Figure 2F*).

To determine whether exosome secretion affects filopodia formation and/or stability, live imaging of B16F1 cells expressing the actin marker tdTomato-F-tractin *Schell et al., 2001* was performed. Analysis of the number of new filopodia formed over time, as well as the lifetime of those filopodia, revealed that Hrs-KD reduces both the formation (*Figure 2G*) and stability (*Figure 2H*) of filopodia (see also *Figure 2—video 1*).

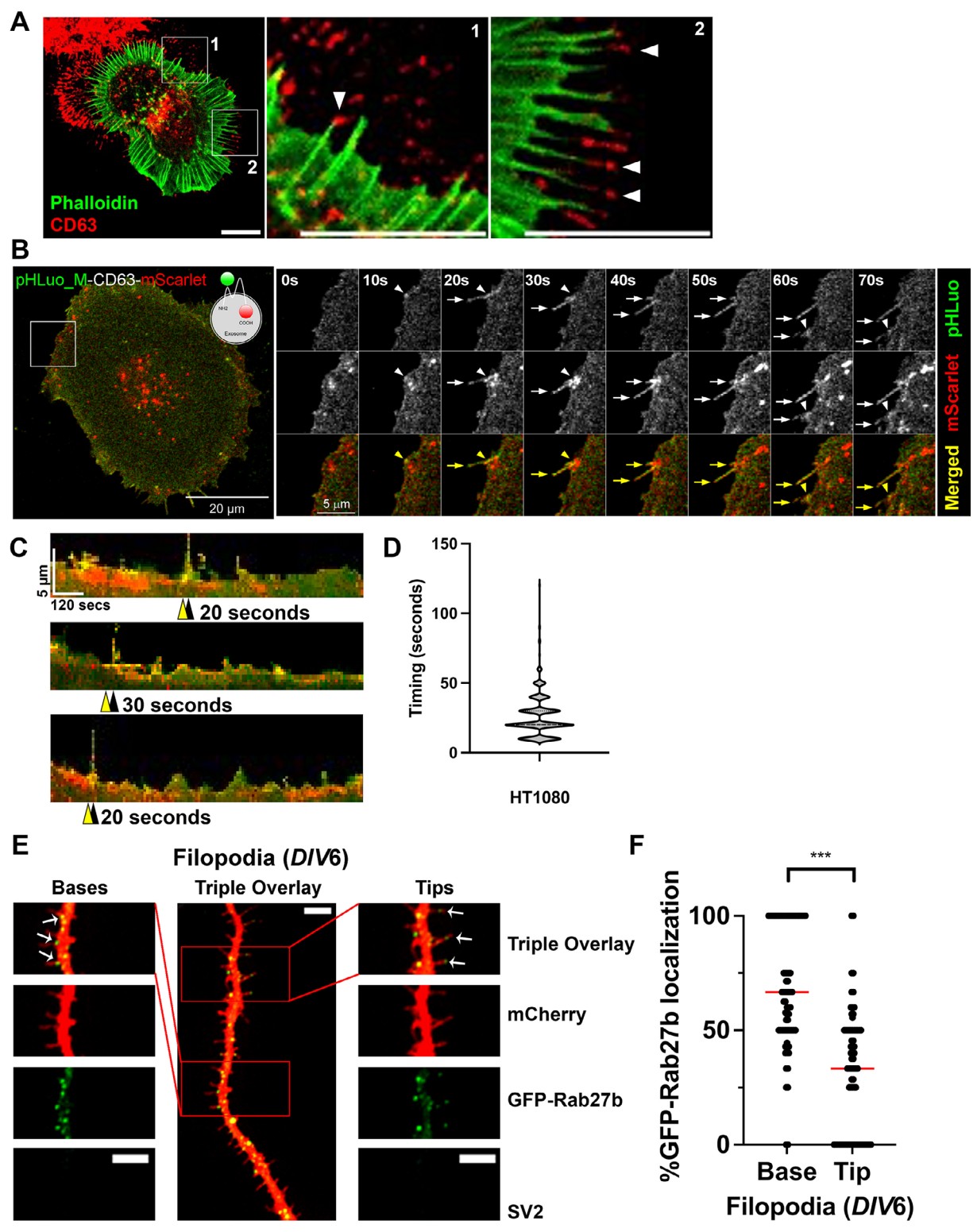

**Figure 1.** Exosome markers localize to the base and tips of filopodia in cancer cells and cortical neurons. (**A**) Representative confocal image of HT1080 cells stained with phalloidin-Alexa fluor 488 and CD63 shown in red. The red channel has been edited using brightness and contrast tools for ease of visibility. Note the localization of the exosome marker CD63 in extracellular deposits and at or near the tips of filopodia (arrowheads). Representative of 20 images. Scale bar is 10 mm in each panel. (**B**) Time series of pHluorin-M153R-CD63-mScarlet movie in HT1080 cells. Yellow arrowheads indicate fusion sites and yellow arrows indicate filopodia. Note a filopodium forming shortly after MVE fusion. (**C**) Representative kymographs showing MVE docking (red), fusion (yellow), and filopodia formation in HT1080 cells. Yellow arrowheads denote MVE fusion events, and black arrowheads denote the

*Figure 1 continued*

formation of a filopodium. Each pixel is 10 s x 0.2857 mm. (**D**) Quantification of the time elapsed between MVE fusion and filopodia formation. n=420 kymographs from 46 cells from three independent experiments (biological replicates). (**E**) Primary cortical neurons were co-transfected with GFP-Rab27b (green) and mCherry as a filler to visualize filopodia (red) on *DIV*5 and fixed for imaging on *DIV*6. SV2 negative staining (no signal) identifies these structures as filopodia instead of dendritic spines. Arrows in merged images indicate localization of GFP-Rab27b to tips and bases of filopodia. Scale bars = 5 µm. (**F**) Percent GFP-Rab27b localization to tips and bases of filopodia in 70 individual cortical neurons from three independent experiments (biological replicates). Red line indicates the median. Error bars, SEM. ns, not significant; * $p<0.05$; ** $p<0.01$; *** $p<0.001$.

The online version of this article includes the following video for figure 1:

**Figure 1—video 1.** Visualization of exosome secretion and filopodia formation dynamics.

https://elifesciences.org/articles/101673/figures#fig1video1

We quantified filopodia in tumor cells as 'filopodia per cell area' to match with corresponding primary neuron data (*Figure 3*), in which we had to plot filopodia per 100 µm length to account for the fact that the entire neuron body was too large to fit in a field of view. To address concerns about this graphing choice, we have included the data presented in *Figure 2* plotted as filopodia per cell (*Figure 2—figure supplement 2G–J*). The data are highly similar, with the one exception being the de novo filopodia in shHrs-2 cells trending the same but not being statistically significant (*Figure 2—figure supplement 2J*). It is possible that this data set is no longer significantly different than the control cells because the cell area of the shHrs-2 cells is slightly higher than that of control cells, although not significantly different.

## Exosomes promote filopodia and synapse formation in neurons

In neurons, filopodia formation is critical for the subsequent development of synapses as filopodia mature into postsynaptic specializations called dendritic spines. In primary rat neuron cultures, numerous filopodia are present on dendritic shafts at day in vitro (*DIV*)6, while dendritic spines are predominant around *DIV*12. To test whether exosomes control filopodia in neurons, we altered the expression of exosome regulators in primary rat hippocampal and cortical neurons. Both Rab27a and Rab27b regulate exosome secretion by respectively promoting MVE docking with the PM and regulating the anterograde movement of MVEs to the PM (*Ostrowski et al., 2010*). However, in some cell types, one form of Rab27 is expressed more highly than the other, suggesting some possible redundant functions (*Barral et al., 2002*). Since Rab27b is the predominant form of Rab27 protein present in the brain (*Zhao et al., 2002*; *Ramalho et al., 2001*), we overexpressed Rab27b in primary neurons. Cortical neurons overexpressing GFP-Rab27b exhibited a significant increase in the number of filopodia examined at *DIV*6, quantitated as thin protrusions that were negative for the synapse marker SV2 (*Figure 3A and B*). This increase in filopodia density translated into an increased number of SV2-positive dendritic spines and synapses at DIV12 (*Figure 3C and D*). A similar increase in filopodia, spine, and synapse density occurred in primary hippocampal neurons upon GFP-Rab27b expression (*Figure 3—figure supplement 1A and B*).

To inhibit exosome secretion in primary cortical and hippocampal neurons, we knocked down Hrs or Rab27b by transient transfection of two different shRNAs along with mCerulean at least 48 hr prior to immunostaining. For both genes, shRNA-transfected neurons (recognized by mCerulean co-transfection) exhibited 40–60% reduction in protein expression analyzed by fluorescence intensity, compared to cells transfected with nontargeting shRNA (NTshRNA, *Figure 3—figure supplement 2A–D*). Analysis of immunostained cells revealed that loss of either Rab27b or Hrs in KD neurons led to a significant reduction in filopodia, spines, and synapse density compared to NTshRNA controls (*Figure 3E-H*, *Figure 3—figure supplement 1C-F*). To further confirm that the effect of gene KD on filopodia density was due to loss of exosome release, we performed rescue experiments by treating Rab27b- or Hrs-KD primary cortical neurons for 24 hr with SEVs isolated by differential centrifugation from *DIV*9 primary cortical neurons (*Figure 3—figure supplement 2E–G*). LEVs were not tested due to the low recovery of LEVs from neuronal conditioned media. The dose of 200 SEVs per neuron was chosen based on the estimated SEV secretion rate from primary cortical neurons over the 24 hr time period of the assay. In the Rab27b-KD condition, SEV treatment fully rescued the filopodia number defects of untreated KD controls (*Figure 3I*, *Figure 3—figure supplement 2H*). For the Hrs-KD condition, there was partial rescue in filopodia density upon SEV treatment (*Figure 3J*, *Figure 3—figure*

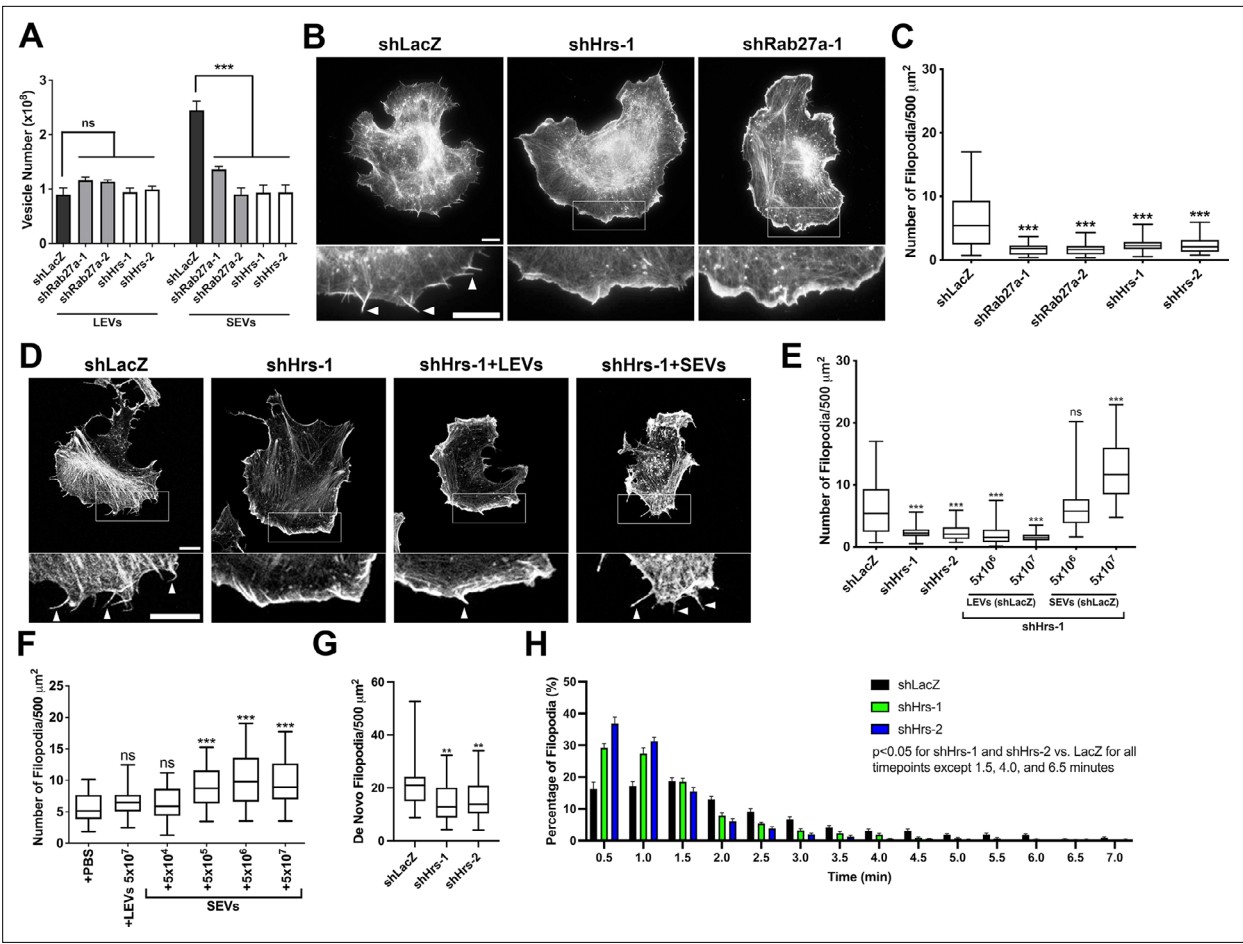

**Figure 2.** Exosomes promote filopodia formation and stability in cancer cells. (**A**) EVs secreted from equal numbers of control (shLacZ), Rab27a-KD, and Hrs-KD B16F1 cells over 48 hr were quantified using NanoSight particle tracking analysis (N=3 biological replicates). (**B**) Representative images of control (shLacZ), Hrs-KD, and Rab27a-KD B16F1 cells stained with rhodamine-phalloidin. Arrowheads show examples of filopodia. Images have been edited with brightness and contrast for ease of visibility. (**C**) Quantification of filopodia from images as in B (≥27 total cells per condition from three biological replicates). Filopodia number per 500 mm² cell area. (**D**) Representative images of filopodia in B16F1 control (shLacZ) and exosome-depleted (shHrs) cells treated for 18 hr with LEVs or SEVs isolated from control cells. Arrowheads show examples of filopodia. Images have been edited with brightness and contrast for ease of visibility. (**E**) Quantification of filopodia from images as in D (≥20 total cells per condition from three biological replicates). (**F**) Filopodia number in B16F1 shScr cells treated with indicated numbers of purified LEVs or SEVs, for 18 hr (≥25 total cells per condition from three biological replicates). (**G**) Control (shLacZ) and exosome-depleted (shHrs) B16F1 cells were transiently transfected with tdTomato-F-tractin to visualize filopodia formation. Live images were taken every 30 s for 15 min and newly formed filopodia were counted at each time point. Only filopodia that form and fully retract during the duration of the video were quantified. (≥20 total cells per type per biological replicate, from three biological replicates). (**H**) Lifetime of newly formed filopodia from G. Lifetime is defined as the time from first formation of the filopodia to full retraction. Bars represent mean and error bars are SEM. Scale bars in wide field and zoom insets = 10 mm. Error bars, SEM. ns, not significant; * p<0.05; ** p<0.01; *** p<0.001.

The online version of this article includes the following video, source data, and figure supplement(s) for figure 2:

**Figure supplement 1.** Characterization of B16F1 cells and EV preparations.

**Figure supplement 1—source data 1.** PDF file containing the original western blots from *Figure 2—figure supplement 1A*, indicating the relevant bands.

**Figure supplement 1—source data 2.** Original files for western blot analysis displayed in *Figure 2—figure supplement 1A*.

**Figure supplement 1—source data 3.** PDF file containing the original western blots from *Figure 2—figure supplement 1B*, indicating the relevant bands.

**Figure supplement 1—source data 4.** Original files for western blot analysis displayed in *Figure 2—figure supplement 1B*.

**Figure supplement 1—source data 5.** PDF file containing the original western blots from *Figure 2—figure supplement 1D*, indicating the relevant bands.

**Figure supplement 1—source data 6.** Original files for western blot analysis displayed in *Figure 2—figure supplement 1D*.

*Figure 2 continued on next page*

*Figure 2 continued*

**Figure supplement 2.** Characterization of HT1080 cells and EV preparations and comparison of graphing methods.

**Figure supplement 2—source data 1.** PDF file containing the original western blots from *Figure 2—figure supplement 2A*, indicating the relevant bands.

**Figure supplement 2—source data 2.** Original files for western blot analysis displayed in *Figure 2—figure supplement 2A*.

**Figure 2—video 1.** Live imaging of filopodia in B16F1 control and shHrs cells.

https://elifesciences.org/articles/101673/figures#fig2video1

*supplement 2H*). These data indicate that - similar to cancer cells - endogenous exosomes secreted by neurons promote filopodia formation.

## Endoglin is an SEV cargo that promotes filopodia formation in cancer cells

EVs carry multiple bioactive protein cargoes. Because only SEVs but not LEVs were able to rescue or induce filopodia numbers in B16F1 melanoma cells (*Figure 2*), we identified unique SEV cargoes by running SEV and LEV lysates on an SDS-PAGE gel, staining it with colloidal Coomassie, extracting four bands that were enriched in the SEV lysates (*Figure 4A*, arrowheads), and performing proteomic analysis of trypsin digests of the extracted protein bands. Our approach was validated by the presence of typical exosomal/SEV proteins CD63, LAMP1, and the ESCRT-1 protein Multivesicular Body Subunit 12B (MVB12B) in one of the unique bands analyzed from the SEV lane (*Figure 4A*, *Supplementary file 1*). In SEV lane bands 1 and 2, we identified the TGF-β co-receptor endoglin (*Figure 4A*). A full list of proteins identified in the SEV-enriched bands is shown in *Supplementary file 1*. Western blot analysis confirmed that endoglin is enriched in SEVs compared to LEVs (*Figure 4B*).

Endoglin is a TGF-β co-receptor that regulates multiple processes relevant to motility and filopodia, including cellular signaling, adhesion, and cytoskeletal regulation (*Cheifetz et al., 1992*; *Nassiri et al., 2011*; *Rossi et al., 2019*; *Schoonderwoerd et al., 2020*). To test whether endoglin regulates filopodia formation, we stably expressed endoglin-targeting shRNAs in B16F1 melanoma cells and confirmed that endoglin levels are reduced in both SEVs and cells (*Figure 4C*). Compared with control cells, stable endoglin-KD cells exhibit a reduction in filopodia number (*Figure 4D and E*) but no significant effect on SEV release from cells (*Figure 4—figure supplement 1A and B*). Likewise, transient KD of endoglin in B16F1 cells also leads to a significant reduction in filopodia numbers (*Figure 4—figure supplement 1C and D*). Whereas the addition of LEVs to stable endoglin-KD cells has little effect on filopodia numbers, treatment with control SEVs fully rescues the filopodia number defects of endoglin-KD cells (*Figure 4D and E*). Consistent with endoglin being a key SEV cargo that controls filopodia, add-back of endoglin-KD SEVs does not rescue the filopodia defects of endoglin-KD cells (*Figure 4D and E*). Furthermore, control - but not endoglin-KD - SEVs also rescue filopodia defects of exosome-inhibited B16F1 Hrs-KD cells (*Figure 4F*, *Figure 4—figure supplement 1E*). To determine whether endoglin selectively controls filopodia formation or stability, we performed live imaging of endoglin-KD B16F1 cells. We found a specific defect of endoglin-KD cells in de novo filopodia formation (*Figure 4G*) with no change in filopodia lifetime (*Figure 4H*, see also *Figure 4—video 1*). These data suggest that endoglin specifically affects filopodia formation, whereas additional exosome cargoes may affect filopodia lifetime, since Hrs-KD cells had a defect in both filopodia formation and lifetime (*Figure 2G and H*). Finally, we assessed the role of endoglin in filopodia formation in HT1080 fibrosarcoma cells. Similar to B16F1 cells, KD of endoglin in HT1080 cells reduced filopodia numbers (*Figure 4—figure supplement 2A–E*). These data suggest that endoglin and/or endoglin-regulated cargoes carried by exosomes have important functions in controlling filopodia dynamics.

## Endoglin promotes cancer cell metastasis and cell motility

We previously observed that exosome secretion is critical for metastasis and directional migration of cancer cells in chick embryos (*Sung et al., 2015*; *Sung and Weaver, 2017*). Since filopodia have been implicated in both of these behaviors (*Mattila and Lappalainen, 2008*; *Arjonen et al., 2011*), we tested the role of endoglin in promoting cancer metastasis in this model. The chick embryo chorioallantoic membrane (CAM) model has been established as an effective way to study experimental metastasis by intravenous injection of tumor cells (*Palmer et al., 2011*). To seed tumor cells at metastatic

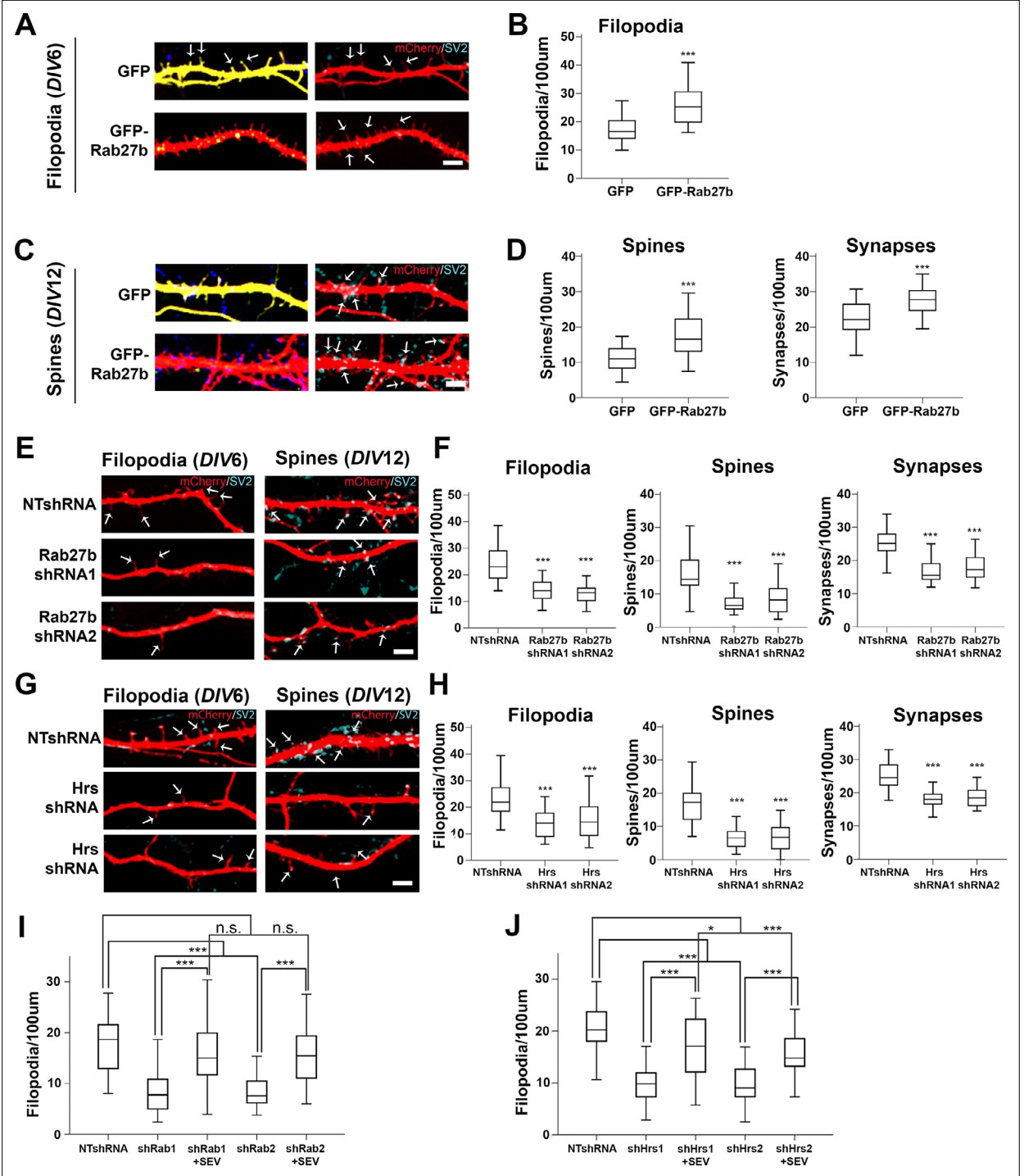

**Figure 3.** Exosomes promote filopodia, spine, and synapse formation in cortical neurons. Primary cortical neurons were co-transfected with plasmids for expression or inhibition of exosome regulatory molecules and mCherry (red) as a cytoplasmic filler to image neuronal protrusions, then fixed and immunostained with SV2 (pseudo-colored in cyan) to visualize synapses. Filopodia were identified as thin SV2-negative, mCherry-positive protrusions. Spines were identified as dendritic protrusions that co-localize with SV2. Synapses were identified as SV2-positive puncta present on both dendritic protrusions and dendritic shafts. (**A, C**) Representative images of primary rat cortical neurons co-transfected at DIV6 when filopodia typically form (**A**) or DIV12 when synapses typically form (**C**) with GFP or GFP-Rab27b (green, left images) and mCherry (red) and co-stained with SV2 (blue, right images). (**B, D**) Quantification of filopodia (DIV6), spine and synapse density (DIV12) from images as in A and C. (**E–G**) Images and analysis from neurons transfected with control shRNA (NTshRNA) or shRNAs against Rab27b (**E**) or Hrs (**G**) and immunostained for SV2. Quantification of filopodia (DIV6), spine and synapse density (DIV12) for Rab27b-KD (**F**) or Hrs-KD (**H**) neurons. (**I, J**) Rescue experiments. Control and KD neurons (as indicated) expressing shRNAs and mCherry were treated with purified SEVs on DIV5 for 24 hr at a dose of 200 EVs per neuron, then fixed and stained for SV2 at DIV6. Filopodia,

*Figure 3 continued on next page*

*Figure 3 continued*

spine, and synapse density were quantified from more than 30 primary or secondary dendritic shafts from three independent experiments (biological replicates) for each condition. Scale bars = 5 mm. Error bars, SEM. *p<0.05, **p<0.01, ***p<0.001.

The online version of this article includes the following source data and figure supplement(s) for figure 3:

**Figure supplement 1.** Exosome secretion enhances filopodia, spine, and synapse numbers in primary hippocampal neurons.

**Figure supplement 2.** Analysis of exosome gene knockdown and SEV rescue in cortical neurons.

**Figure supplement 2—source data 1.** PDF file containing the original western blots from *Figure 3—figure supplement 2F*, indicating the relevant bands.

**Figure supplement 2—source data 2.** Original files for western blot analysis displayed in *Figure 3—figure supplement 2F*.

sites, we injected fluorescently labeled control and endoglin KD HT1080 cells intravascularly. After 4 days, we harvested and imaged the chick CAM to visualize metastatic cells and colonies (*Figure 5A*). Many cancer cells and colonies were present in the CAM from shScr control cells. By contrast, CAMs from chick embryos injected with endoglin-KD HT1080 cells had reduced numbers of individual metastatic cancer cells and colonies (*Figure 5B–D*). Furthermore, there was an increased number of large colonies, which is a phenotype typically observed in cells with defects in migration away from growing colonies (*Figure 5E*; *Sung et al., 2015*; *Zijlstra et al., 2008*; *Palmer et al., 2014*). This phenotype is similar to that which we previously observed with Rab27a-KD HT1080 cells (*Sung et al., 2015*).

To determine whether alterations in exosomes were responsible for the effect of endoglin in vivo, we tested whether coinjection of SEVs with the cells could rescue the metastatic defects of endoglin-KD cells. The number of SEVs that were injected with cells was determined using the estimated SEV secretion rate calculated from nanoparticle tracking analysis (NTA) data of 9.28 SEVs per cell per hour for HT1080 control cells. Since the chick CAM experiment was conducted over a period of 96 hr, and each injection contained 100,000 cells, we used 89 million SEVs per injection. Indeed, coinjection of SEVs led to a full rescue of the endoglin-KD phenotype, both increasing the number of metastatic colonies and decreasing the number of large colonies (*Figure 5B–E*).

Given the known role of filopodia in directed cell migration (*Jacquemet et al., 2015*) and the presence of large, potentially nonmotile, colonies formed by endoglin-KD cells in the chick CAM, we hypothesized that endoglin-KD cells would have a defect in cellular migration. To test this possibility, control and endoglin-KD HT1080 cells were allowed to form spheroids and then embedded in 3D type I collagen. Movies were taken to observe and quantitate cell migration away from the spheroids (*Figure 5—video 1*). Comparison of the spheroid area increase after 8 hr, including migration of individual cells away from the spheroids, revealed that endoglin-KD cells indeed have a defect in 3D cell migration compared to controls. This migratory defect was rescued by re-expression of WT endoglin in the KD cells, confirming that it was not due to an off-target effect of the shRNA (*Figure 5F*, *Figure 5—video 1*).

## Endoglin controls levels of the filopodia regulator THSD7A in cancer cell SEVs

Endoglin is not expressed in neurons (*Matsubara et al., 2000*); therefore, it could not be a universal filopodia regulator. In addition, as a TGFβ coreceptor, it is difficult to imagine how endoglin as an EV cargo could directly induce filopodia formation by recipient cells. However, endoglin interacts with numerous proteins, including TGFβ receptors Alk1 and Alk5, TGFβ family ligand BMP9, and α5β1 integrin, and could potentially alter their sorting into exosomes (*Cheifetz et al., 1992*; *Schoonderwoerd et al., 2020*; *Barbara et al., 1999*; *Jonker, 2014*; *Tian et al., 2012*). Thus, we hypothesized that endoglin could promote filopodia formation by carrying a key filopodia-inducing cargo into exosomes. Surprisingly, Western blot analysis of control and endoglin-KD SEVs did not reveal reductions in several candidate endoglin binding partners in KD SEVs, including TGF-β1, Alk1, or β1 integrin (*Figure 6—figure supplement 1A*). As BMP9 is the only TGFβ family member reported to directly bind to endoglin (*Castonguay et al., 2011*), we also tested whether BMP9 could affect filopodia formation. Contrary to expectation, purified BMP9 did not rescue the defect in filopodia formation by endoglin-KD cells and instead decreased filopodia numbers in control shScr cells (*Figure 6—figure supplement 1B*). We also tested whether the addition of TGF-β1 or coating the culture surfaces with fibronectin might rescue the filopodial defect of endoglin-KD cells. While both of those ligands

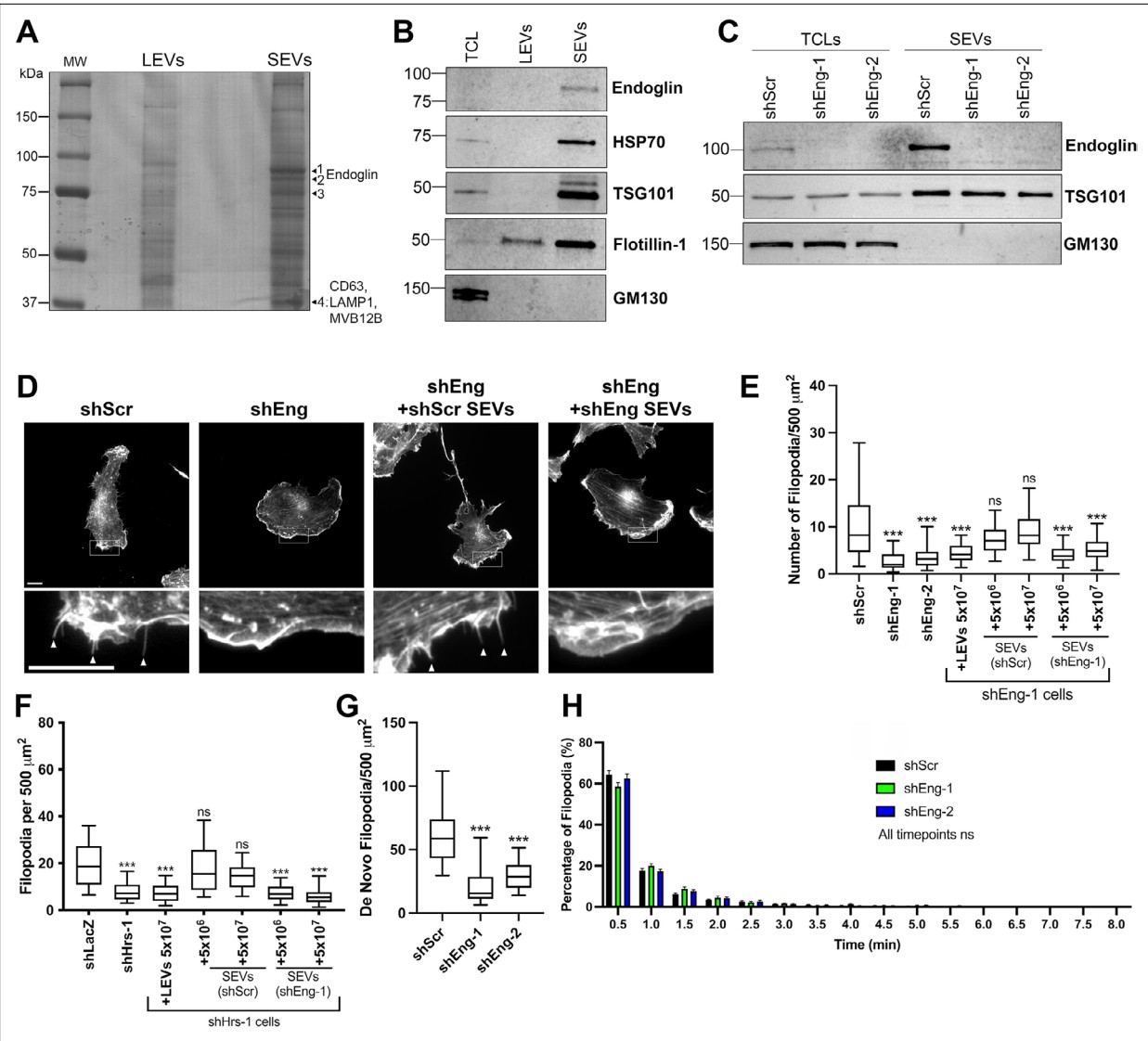

**Figure 4.** Endoglin is an SEV-enriched cargo that promotes filopodia formation. (**A**) Purified LEVs and SEVs were run on a colloidal blue-stained gel. Four arrows denote SEV bands that were cut and submitted for proteomics, along with notable proteins identified (see ***Supplementary file 1*** for the full proteomics results). (**B**) B16F1 total cell lysate (TCL), LEVs, and density gradient purified SEVs were run on an SDS-PAGE gel and probed by western blot for endoglin, and EV positive (HSP70, TSG101, flotillin-1, and CD63) and negative (GM130) markers. (**C**) Total cell lysate (TCL) and small EVs (SEVs) from endoglin-KD (shEng) and control (shScr) B16F1 cells were run on an SDS-PAGE gel and probed by western blot for endoglin, EV marker TSG101, and EV-negative marker GM130. (**D**) Representative images from control (shScr) or endoglin-KD (shEng) B16F1 cell lines incubated for 18 hr with no EVs (left panels), or with SEVs purified from control (+shScr SEVs) or shEng cell lines (+shEng SEVs) (right panels). Arrowheads indicate example filopodia. Scale bar = 10 mm. (**E**) Quantification of filopodia in control (shScr) and endoglin knockdown (shEng) cells treated with the indicated number of LEVs or SEVs for 18 hr (≥20 cells per condition per biological replicate, from three biological replicates). (**F**) Filopodia number in B16F1 control (shLacZ) or exosome-depleted (shHrs) cells treated with indicated numbers of LEVs, control (shScr) SEVs, or endoglin-KD (shEng1) SEVs for 18 hr. ≥20 cells per condition per biological replicate, from three biological replicates. Representative images for this experiment are shown in ***Figure 5E***. (**G, H**) B16F1 cells were transfected with tdTomato-F-Tractin and imaged live every 30 s for 15 min. Only filopodia that form and fully retract during the duration of each video were quantified. (**G**) De novo filopodia formation. (**H**) Filopodia lifetime, defined as the time from initial filopodia formation to full retraction. Bars represent mean and error bars are SEM. (≥25 total cells per type per biological replicate, from three biological replicates) ns, not significant; * p<0.05; ** p<0.01; *** p<0.001.

The online version of this article includes the following video, source data, and figure supplement(s) for figure 4:

**Source data 1.** PDF file containing original blot for ***Figure 4A***, indicating the relevant bands.

**Source data 2.** Original file for Coomassie blue gel displayed in ***Figure 4A***.

**Source data 3.** PDF file containing the original western blots from ***Figure 4B***, indicating the relevant bands.

*Figure 4 continued on next page*

*Figure 4 continued*

**Source data 4.** Original files for western blot analysis displayed in *Figure 4B*.

**Source data 5.** PDF file containing the original western blots from *Figure 4C*, indicating the relevant bands.

**Source data 6.** Original files for western blot analysis displayed in *Figure 4C*.

**Figure supplement 1.** KD of endoglin affects filopodia numbers but not EV numbers in B16F1 melanoma cells.

**Figure supplement 1—source data 1.** PDF file containing the original western blots from *Figure 4—figure supplement 1C*, indicating the relevant bands.

**Figure supplement 1—source data 2.** Original files for western blot analysis displayed in *Figure 4—figure supplement 1C*.

**Figure supplement 2.** HT1080 Endoglin-KD cells have reduced filopodia.

**Figure supplement 2—source data 1.** PDF file containing the original western blots from *Figure 4—figure supplement 2A*, indicating the relevant bands.

**Figure supplement 2—source data 2.** Original files for western blot analysis displayed in *Figure 4—figure supplement 2A*.

**Figure 4—video 1.** Live imaging of filopodia in B16F1 control and shEng cells.

https://elifesciences.org/articles/101673/figures#fig4video1

---

slightly enhanced the number of filopodia in control cells, there was no effect on endoglin-KD cells (*Figure 6—figure supplement 1C and D*).

To determine whether endoglin may regulate the transport of any novel proteins into exosomes, we performed a quantitative proteomic comparison between SEVs purified from control and endoglin-KD B16F1 cells. Equal amounts of protein extracted from the SEVs were subjected to isobaric tagging for relative and absolute quantitation (iTRAQ) and mass spectrometry analysis (*Mertins et al., 2012*; *Ross et al., 2004*). Consistent with our western blot results (Fig. S7A), the proteomics data showed no significant differences in the expression of any integrins or TGFβ family ligands or receptors (*Supplementary file 2*). Analysis of the data further revealed only two proteins that were significantly lower in both endoglin KD1 and KD2 samples compared to control samples (*Supplementary file 2*), endoglin and the extracellular matrix (ECM) protein thrombospondin type 1 domain containing 7 a protein (THSD7A).

THSD7A is a transmembrane protein that is known to be expressed in endothelial cells, neurons, and kidney podocytes (*Herwig et al., 2019*; *Kuo et al., 2011*; *Liu et al., 2016*; *Wang et al., 2010*). Interestingly, although little is known about this protein, a soluble form of THSD7A was reported to promote filopodia formation in endothelial cells (*Kuo et al., 2011*). Additionally, THSD7A has been shown to be aberrantly expressed in a variety of cancers and autoantibodies against THSD7A are a frequent cause of a paraneoplastic autoimmune kidney disease, secondary membranous nephropathy (*Hou et al., 2017*; *Stahl et al., 2017*; *Wang et al., 2019*). Autoantibodies against THSD7A are also a cause of idiopathic (primary) membranous nephropathy (*Tomas et al., 2014*).

To validate our finding that THSD7A levels are downregulated in SEVs from endoglin-KD cancer cells, we performed Western blot analysis of control and endoglin-KD SEVs from B16F1 and HT1080 cells (*Figure 6A, B*). Detection of THSD7A in B16F1 SEVs necessitated using native gels and thus endoglin and SEV marker CD63 ran at a significantly higher molecular weight (*Figure 6A*). Indeed, THSD7A was present on control SEVs and was reduced on endoglin-KD SEVs purified from either cell line. THSD7A was also present on SEVs isolated from primary cortical neurons (*Figure 6C*).

To test whether THSD7A regulates filopodia formation in our system, we overexpressed THSD7A tagged with mScarlet fluorescent protein in HT1080 cells. THSD7A-mScarlet localized to the tips of filopodia as well as in extracellular deposits similar to those we have previously observed with exosomes (*Figure 6*, *Sung et al., 2015*; *Sung et al., 2020*). Notably, expression of THSD7A-mScarlet increased filopodia numbers in HT1080 cells (*Figure 6D*, graph). To further test the role of THSD7A in filopodia formation, we knocked down THSD7A in HT1080 cells using 3 different targeting shRNAs (*Figure 6E*). Analysis of filopodia numbers revealed a significant reduction in THSD7A-KD cells (*Figure 6E*). To determine the role of THSD7A in filopodia regulation by endoglin, we plated control and endoglin-KD cells on coverslips coated with various concentrations of recombinant THSD7A protein. Indeed, recombinant human THSD7A fully rescued the filopodia defects of endoglin-KD for both HT1080 and B16F1 melanoma cells (*Figure 6F*).

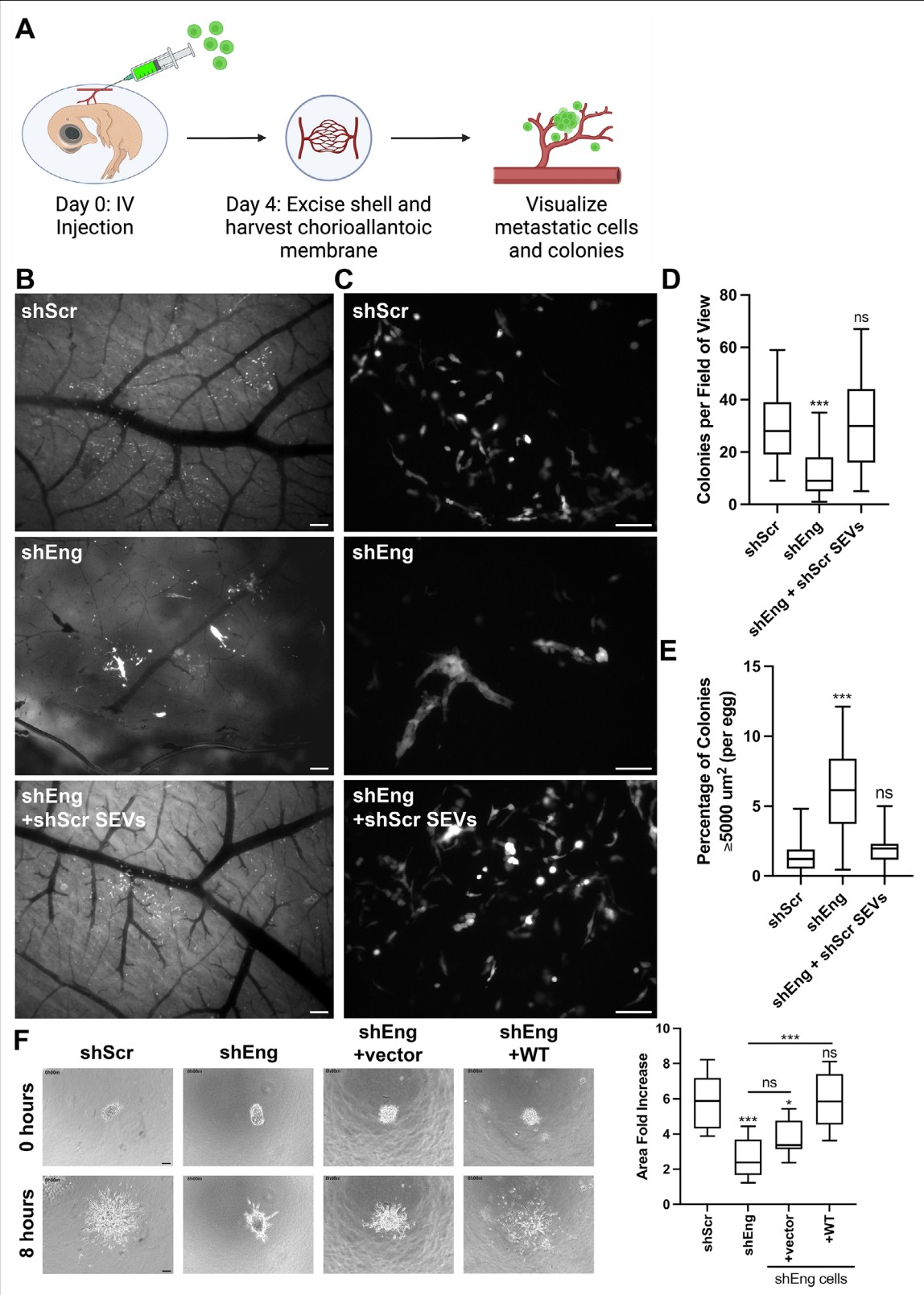

**Figure 5.** Exosomal endoglin controls motility and metastatic colony formation. (**A**) Cartoon diagram of metastatic colony assay in avian embryos. On day 0, fluorescent HT1080 cells were injected (100,000 cells per egg) into the vein of the chicken embryo. On day 4, the egg was opened, the embryo was sacrificed, and a circular tool was used to punch holes through the shell. The chorioallantoic membrane (CAM) was peeled away from the shell, placed on a glass slide with a coverslip, and immediately imaged. The cartoon was created using BioRender.com. (**B**) Representative low power

*Figure 5 continued on next page*

*Figure 5 continued*

wide field images of colony formation in the CAM. Scale bar = 200 mm. (**C**) Representative high-power wide field images of colony formation in the CAM. Scale bar = 100 mm. (**D, E**) Quantification of CAM colony number (**D**) and size (**E**) from high-power images as in C. 4–7 eggs were harvested per replicate for each condition for three biological replicates. (**D**) Colony number is graphed per field of view using 25–30 fields of view per egg. (**E**) Quantification of the percent of large (≥5000 mm$^2$) colonies formed by control and shEng HT1080 cells. (**F**) 3D invasion in collagen. HT1080 cell spheroids were seeded in collagen gels and imaged for 8 hr. Invasion is quantified as fold area increase in the size of each spheroid over 8 hr. Scale bar = 100 mm. Error bars, SEM. ns, not significant; *p<0.05; ** p<0.01; *** p<0.001.

The online version of this article includes the following video for figure 5:

**Figure 5—video 1.** Endoglin controls 3D migration in type I collagen.

https://elifesciences.org/articles/101673/figures#fig5video1

Our typical filopodia analysis is performed after culturing cells for 48 hr on coverslips. To determine whether THSD7A can boost filopodia formation in a short time period as would be expected for a direct regulator, we plated cells on coverslips coated with poly-D-Lysine or recombinant THSD7A for 15 min-2 hr and found that recombinant THSD7A fully rescued the filopodia defect of endoglin-KD cells at the shortest timepoint of 15 min (*Figure 6—figure supplement 1E*).

Although neurons do not express endoglin (*Matsubara et al., 2000*), they do rely on exosomes for filopodia induction (*Figure 3*) and they express THSD7A (*Figure 6C*, *Liu et al., 2016*; *Wang et al., 2011*). To test the role of THSD7A in neuronal filopodia formation, we transiently expressed FLAG-tagged THSD7A in cortical neurons. Immunostaining analysis revealed that neurons expressing FLAG-THSD7A had localization of the THSD7A to filopodia tips and increased numbers of filopodia compared to vector control transfected neurons (*Figure 6G and H*). In addition, plating exosome-inhibited Hrs-KD neurons on recombinant THSD7A-coated coverslips fully rescued filopodia formation in a dose-dependent manner (*Figure 6I*). These data indicate that THSD7A is an important filopodia-inducing SEV cargo in diverse cell types.

## Endoglin regulates THSD7A trafficking into cancer cell exosomes

Based on the observed decreases in THSD7A expression in SEVs from endoglin-KD cancer cells (*Figure 6A and B*) and the role of endoglin as a co-receptor that traffics through endosomes, we hypothesized that endoglin may alter the trafficking of THSD7A to SEVs/exosomes. To test this hypothesis, we first investigated the relative abundance of THSD7A in control and endoglin-KD HT1080 cells and SEVs by Western blot analysis. Indeed, lysates of endoglin-KD cells had increased levels of THSD7A compared to control cells, while endoglin KD SEVs had decreased levels of THSD7A (*Figure 7A–C*). These data are consistent with an alteration in THSD7A trafficking in endoglin-KD cells rather than a decrease in the overall protein expression of THSD7A. We performed a rescue experiment, in which endoglin-KD cells were stably transfected with either wild-type endoglin (+WT) or an empty vector control (+vector). Interestingly, we found that the high cellular levels of THSD7A in endoglin-KD cells were decreased with re-expression of WT endoglin (*Figure 7A–C*). These changes were paralleled by increased THSD7A in +WT endoglin rescue SEVs. In parallel, expression of WT endoglin in HT1080 endoglin KD cells also rescues filopodia defects (*Figure 7D*).

To determine whether endoglin might affect the trafficking of THSD7A to MVE, we expressed THSD7A-mScarlet in control and endoglin-KD HT1080 cells and co-stained with an anti-CD63 antibody to label MVEs as well as with phalloidin to visualize the cell boundary (*Figure 7E*). Consistent with less THSD7A being secreted in EVs, there was much less extracellular THSD7A surrounding endoglin-KD cells compared to control cells (*Figure 7E*, Zoom 1). In addition, we observed increased accumulation of THSD7A in CD63-positive MVE in endoglin-KD cells compared to control cells (*Figure 7E*, Zoom 2). Thus, the percentage of intracellular THSD7A colocalized with CD63-positive compartments was increased in KD cells compared with control cells (*Figure 7F*), suggesting increased accumulation in endolysosomal compartments. These data suggest that endoglin alters the endolysosomal trafficking of THSD7A, leading to decreased THSD7A release in exosomes.

## Cdc42 activity is required for filopodia induction by THSD7A and endoglin

Finally, we sought to identify downstream signaling pathways responsible for modulating filopodia dynamics via exosomal endoglin and THSD7A. To assess any changes in TGFβ signaling, we assayed for

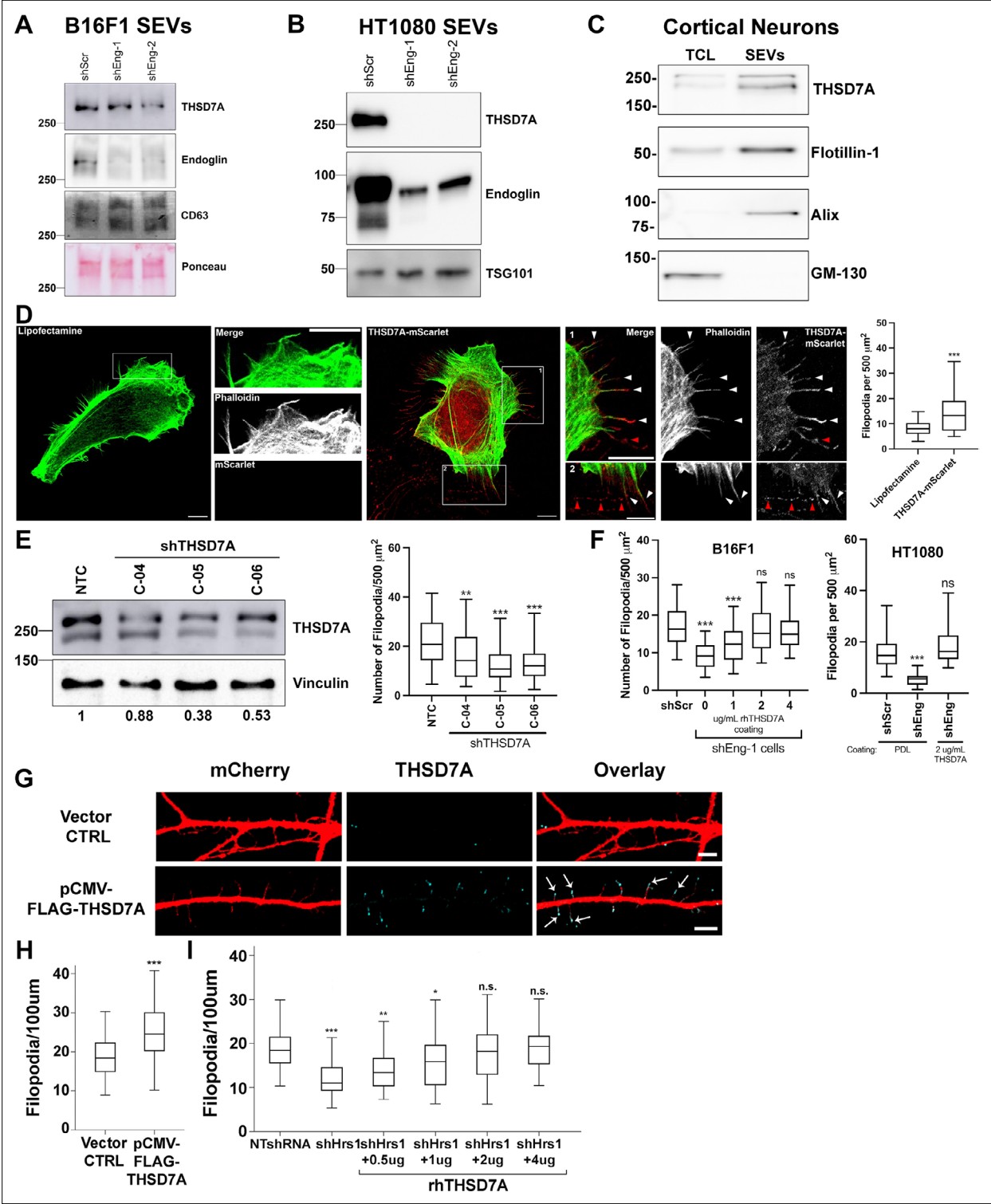

**Figure 6.** Exosomal endoglin promotes filopodia formation through THSD7A. (**A**) Native gel Western blot of B16F1 SEVs. (**B**) Standard western blot of HT1080 SEVs. (**C**) Western blot of cortical neuron total cell lysate (TCL) and SEVs. (**D**) Representative images and quantitation of filopodia number in control (lipofectamine) and THSD7A-mScarlet-transfected HT1080 cells. Arrowheads indicate THSD7A at the ends of filopodia (white arrowheads) or in extracellular deposits (red arrowheads). Scale bars in wide field and zoom insets = 10 mm. (**E**) (Left) Western blot of control shRNA (NTC) and shTHSD7A (C-04, C05, C-06) - expressing HT1080 cell lines. Vinculin is used as a loading control and numbers below the blot indicate normalized THSD7A levels. (Right) Filopodia counts in control and shTHSD7A HT1080 cells. ≥20 cells per condition per biological replicate, from three biological replicates. (**F**) THSD7A coated coverslips rescue filopodia defect in shEng B16F1 and HT1080 cells.≥20 cells per condition per biological replicate, from

*Figure 6 continued on next page*

*Figure 6 continued*

three biological replicates. (**G, H**) Cortical neurons were transfected with a FLAG-THSD7A expression vector (*Kuo et al., 2011*) or vector control, fixed, and stained with an antibody against THSD7A, and imaged by confocal microscopy. (**G**) Representative images. Arrows indicate THSD7A localization to the tips of filopodia. Scale bar = 5 mm. (**H**) Quantification of filopodia in neurons expressing FLAG-THSD7A or control vector. n=42 neurons from three separate experiments (biological replicates). (**I**) Rescue of filopodia numbers in shHrs neurons plated on dishes coated with various concentrations of recombinant human THSD7A, as indicated. Error bars, SEM. ns, not significant; * p<0.05; ** p<0.01; *** p<0.001.

The online version of this article includes the following source data and figure supplement(s) for figure 6:

**Source data 1.** PDF file containing the original western blots and Ponceau stain from *Figure 6A*, indicating the relevant bands.

**Source data 2.** Original files for western blot and Ponceau analysis displayed in *Figure 6A*.

**Source data 3.** PDF file containing the original western blots from *Figure 6B*, indicating the relevant bands.

**Source data 4.** Original files for western blot analysis displayed in *Figure 6B*.

**Source data 5.** PDF file containing the original western blots from *Figure 6C*, indicating the relevant bands.

**Source data 6.** Original files for western blot analysis displayed in *Figure 6C*.

**Source data 7.** PDF file containing the original western blots from *Figure 6E*, indicating the relevant bands.

**Source data 8.** Original files for western blot analysis displayed in *Figure 6E*.

**Figure supplement 1.** Analysis of candidate EV cargoes for rescue of filopodia defects in shEng cells.

**Figure supplement 1—source data 1.** PDF file containing the original western blots from *Figure 6—figure supplement 1A*, indicating the relevant bands.

**Figure supplement 1—source data 2.** Original files for western blot analysis displayed in *Figure 6—figure supplement 1A*.

downstream Smad phosphorylation and did not detect any increases from treatment with rhTHSD7A coating (*Figure 8—figure supplement 1*). Because the small GTPase Cdc42 has been shown to regulate filopodia across diverse cell types (*Blake and Gallop, 2023*; *Nobes and Hall, 1995*), we hypothesized that THSD7A and endoglin might induce filopodia formation via Cdc42 activation. To test this hypothesis, we determined whether filopodia induced by THSD7A could be inhibited with a specific Cdc42 inhibitor or, conversely, whether expressing a constitutively active form of Cdc42 could rescue the filopodia defects of endoglin-KD cells (*Figure 8*). The Q61L mutant of Cdc42 (Cdc42-Q61L) is a dominant active, GTPase-defective, GTP-bound form of Cdc42 (*Ziman et al., 1991*). ML141 has been identified as a non-competitive inhibitor of Cdc42 and effectively inhibits Cdc42-Q61L (*Surviladze et al., 2010*). Consistent with a role for Cdc42 activation in basal filopodia formation by our cells, we found that treatment of control HT1080 cells with the ML141 inhibitor significantly reduces filopodia numbers (*Figure 8*). Likewise, the rescue of filopodia numbers that occurs when endoglin-KD cells are seeded on THSD7A-coated coverslips is ablated when combined with ML141 treatment (*Figure 8*). In addition, the filopodia defect of endoglin-KD cells is rescued by expression of Cdc42-Q61L. The filopodia induction by Cdc42-Q61L in HT1080 endoglin-KD cells was inhibited by ML141, confirming that ML141 acts to inhibit Cdc42 at the doses and times used (*Figure 8*).

## Discussion

Filopodia are adhesive cellular structures that control cell polarization, chemical and physical sensing, and motility. Using a genetic inhibition and rescue approach, we found that autocrine secretion of exosomes controls the number and stability of filopodia in both cancer cells and neurons. The decrease in filopodia in neurons was paralleled by a similar decrease in synapse formation. In cancer cells, we identified endoglin as a key SEV cargo and molecular regulator of this process, controlling filopodia formation, cell migration, and metastasis of cancer cells via SEVs. As endoglin seemed unlikely to directly induce filopodia formation and is not present in neurons, we performed quantitative proteomics of control and endoglin-KD SEVs and identified the transmembrane ECM protein THSD7A as regulated in cancer cell SEVs by endoglin. In neurons, THSD7A was also present on SEVs and regulated filopodia formation. Indeed, purified THSD7A was able to rescue filopodia defects in both endoglin-KD cancer cells and exosome-inhibited neurons. Finally, we found that Cdc42 activity is important for filopodia formation controlled by endoglin and THSD7A. Altogether, we identify THSD7A carried by exosomes as a key controller of filopodia formation in diverse cell types and

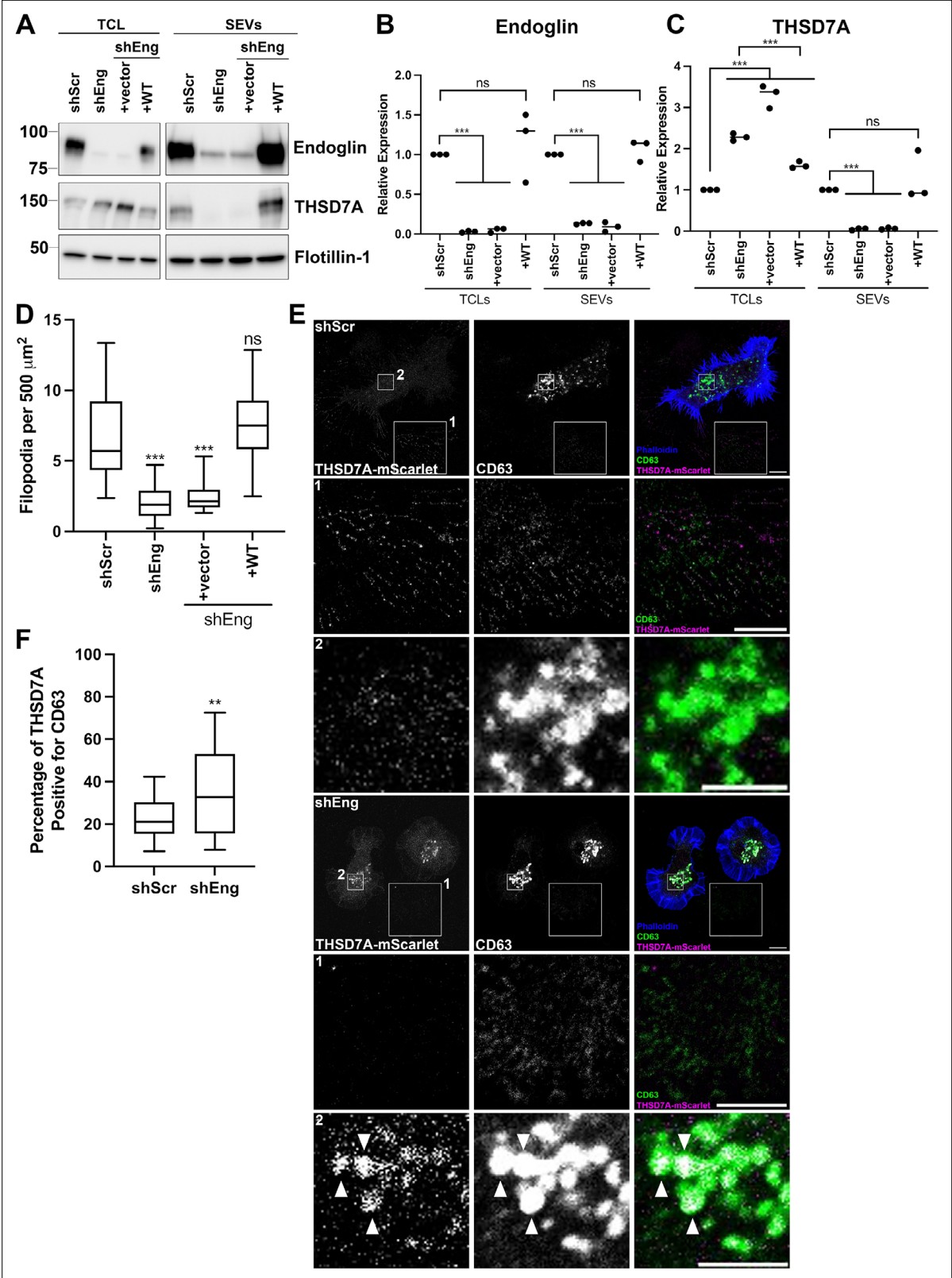

**Figure 7.** Endoglin controls THSD7A trafficking to exosomes. (**A**) Western blot analysis of total cell lysates (TCL) and SEVs from HT1080 control and shEng cells +/-rescue with WT endoglin or control expression vectors. The figure was made from cropped images of membranes to remove irrelevant lanes. (**B**) Quantification of endoglin expression (normalized to flotillin-1 as a loading control, and relative to shScr control) from triplicate Western blots as in A. (**C**) Quantification of THSD7A expression (relative to flotillin-1 as a loading control, and relative to shScr control) from triplicate Western blots

*Figure 7 continued on next page*

*Figure 7 continued*

as in A. (**D**) Quantification of filopodia in HT1080 control cells and shEng cells rescued with WT endoglin expression. N=3, at least 30 total cells per condition. (**E**) Representative confocal images of THSD7A-mScarlet-expressing control and shEng HT1080 cells immunostained for CD63. Box 1 shows extracellular THSD7A and CD63 deposits. Box 2 shows intracellular CD63-positive MVEs. For both boxes, the zoomed images have been adjusted for brightness and contrast (to equivalent levels for control and shEng cells) for easy visualization. Note that the overlap of THSD7A (magenta) and CD63 (green) gives a white signal, pointed out with white arrowheads in the shEng merged image in Zoom 2. Scale bar is 10 mm in wider field view and 5 mm in zoom insets. (**F**) Quantification of colocalization of internal CD63 and mScarlet signals in HT1080 cells from nonadjusted images.≥20 cells per condition per biological replicate, from three biological replicates. Error bars, SEM. ns, not significant; * p<0.05; ** p<0.01; *** p<0.001.

The online version of this article includes the following source data for figure 7:

**Source data 1.** PDF file containing the original western blots from *Figure 7A*, indicating the relevant bands.

**Source data 2.** Original files for western blot analysis displayed in *Figure 7A*.

further identify endoglin as a key regulator of THSD7A secretion in exosomes in cancer cells. These data suggest a new model for filopodia induction via secreted exosomes (*Figure 9*).

Exosomes have been shown to regulate cell migration and invasion in a variety of contexts (*Kriebel et al., 2018*; *Majumdar et al., 2021*; *Sung et al., 2015*; *Sung et al., 2021*; *Sung and Weaver, 2017*; *Sung et al., 2020*). In order for cells to migrate, they must form adhesive and sensing cytoskeletal structures, including lamellipodia, focal complexes, invadopodia, and filopodia. Our finding that exosomes promote filopodia formation adds to our previous findings that exosome secretion promotes the formation of nascent adhesions and invadopodia (*Sung et al., 2015*; *Hoshino et al.,*

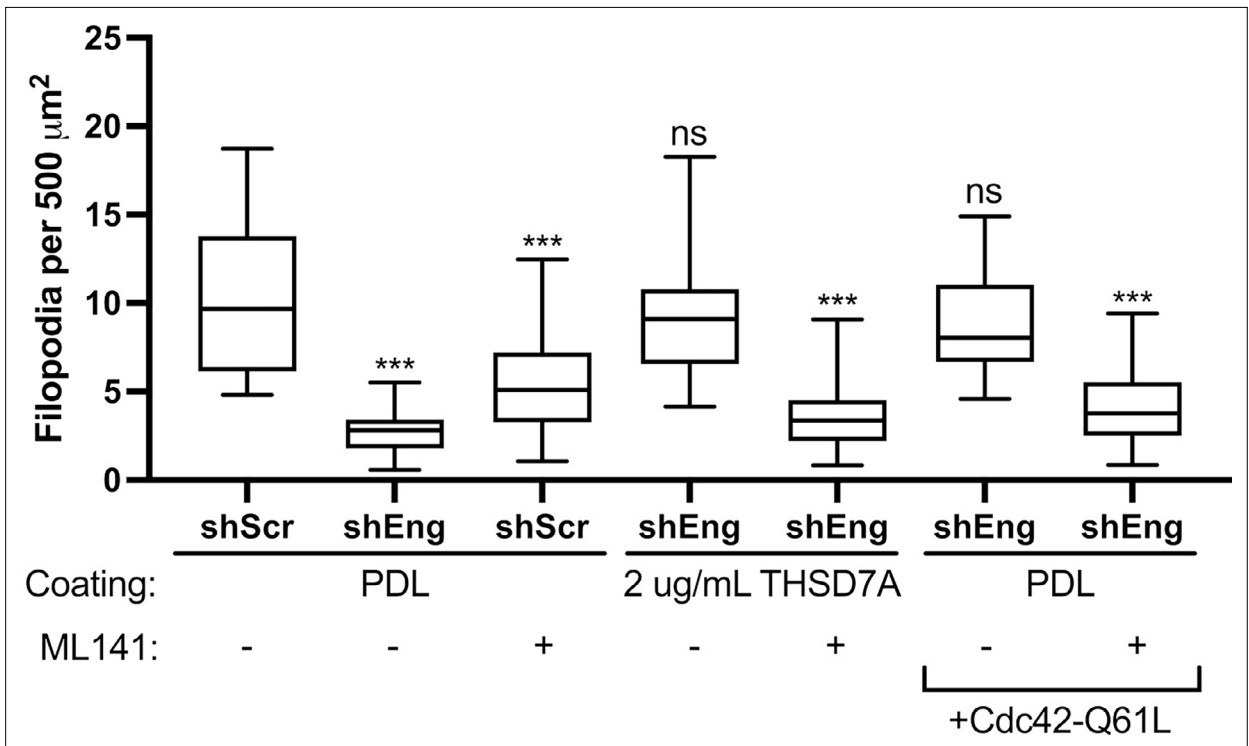

**Figure 8.** Filopodia formation induced by THSD7A depends on Cdc42 activity. Control and endoglin-KD HT1080 cells were plated on coverslips coated with poly-D-lysine (PDL) or THSD7A. In some cases, cells were treated with the Cdc42 inhibitor ML141 (10 µM) or transfected with the dominant active Cdc42 mutant Q61L, as indicated.≥20 cells per condition per biological replicate, from three biological replicates. Error bars, SEM. ns, not significant; * p<0.05; ** p<0.01; *** p<0.001.

The online version of this article includes the following source data and figure supplement(s) for figure 8:

**Figure supplement 1.** Total cell lysates of B16F1 cells were seeded on PDL +/-THSD7 A and treated with or without TGF-b1 and inhibitor.

**Figure supplement 1—source data 1.** PDF file containing the original western blots from *Figure 8—figure supplement 1*, indicating the relevant bands.

**Figure supplement 1—source data 2.** Original files for western blot analysis displayed in *Figure 8—figure supplement 1*.

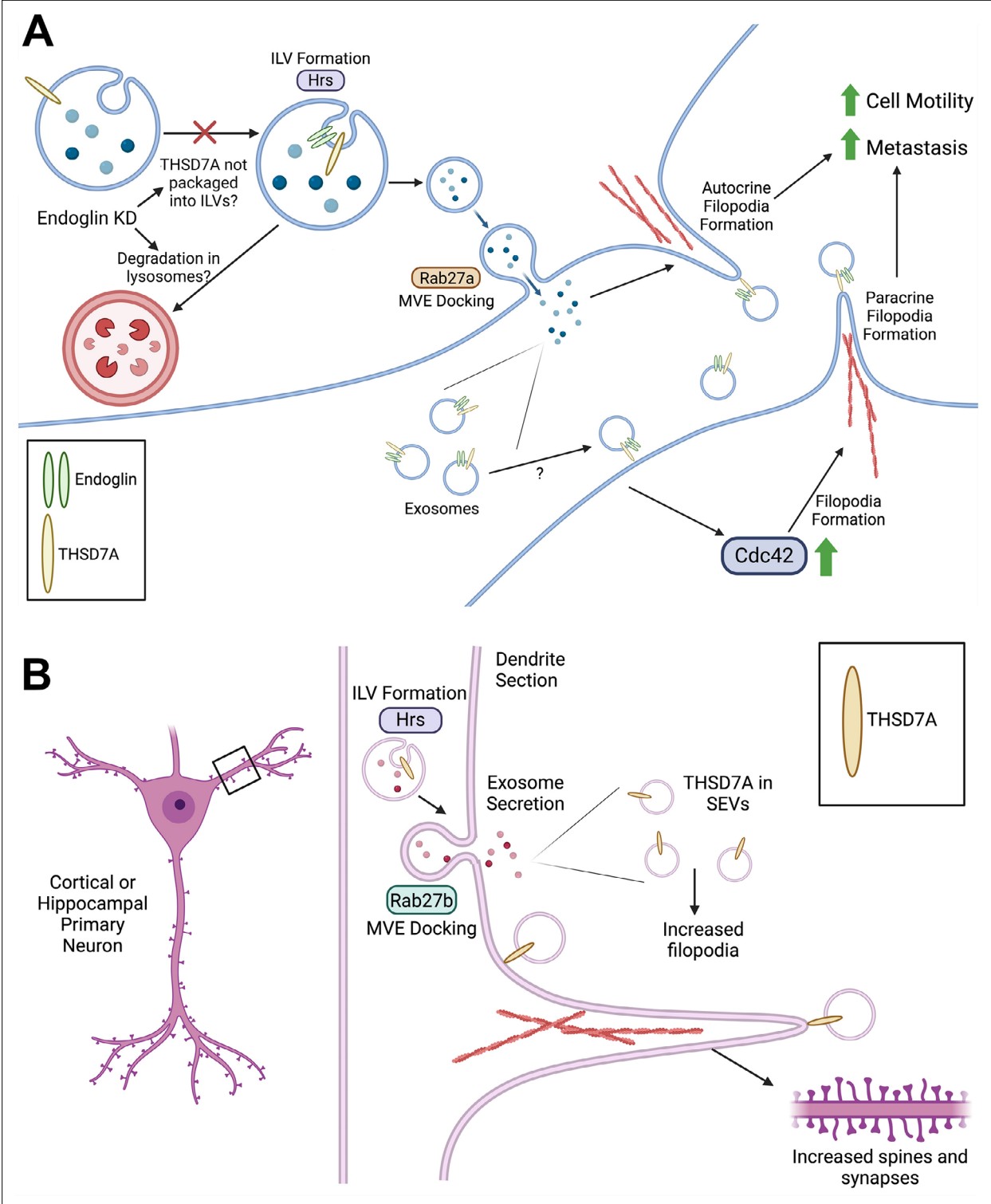

**Figure 9.** Model of exosome-induced filopodia formation. (**A**) In tumor cells, endoglin and THSD7A are trafficked into intralumenal vesicles (ILV) in multivesicular endosomes (MVEs) for secretion. Inhibiting the exosome biogenesis pathway by blocking Hrs or inhibiting MVE docking by blocking Rab27a subsequently reduces exosome secretion and filopodia formation. SEVs carrying THSD7A can induce filopodia on target cells via Cdc42, leading to increased cell motility and metastasis. When endoglin levels are lowered (such as by KD), THSD7A is retained inside cells in CD63-positive endosomes, and its levels in SEVs are greatly decreased. The drop in THSD7A levels in endoglin-KD EVs could be due either to a lack of trafficking into ILVs or, alternatively, enhanced lysosomal degradation of THSD7A-containing MVEs. Given that THSD7A accumulates in CD63-positive endolysosomal compartments in endoglin-KD cells (*Figure 7E and F*), the latter possibility seems more likely. The cartoon was created using BioRender.com. (**B**) In

*Figure 9 continued on next page*

*Figure 9 continued*

primary neurons, exosome biogenesis is controlled by the formation of ILVs by Hrs and MVE docking is controlled by Rab27b. Knockdown of either of these proteins results in reduced formation of filopodia, dendritic spines, and synapses in both cortical and hippocampal neurons. Similar to cancer cells, THSD7A is carried in neuronal SEVs and induces filopodia. The cartoon was created using BioRender.com.

*2013*) and likely stabilizes lamellipodia (*Bryce et al., 2005*; *Sung et al., 2011*; *Sinha et al., 2016*). While some of these activities are likely related, the filopodia activity appears to be distinct. Indeed, while we previously found that fibronectin was a critical exosome cargo driving nascent adhesion formation, lamellipodia stabilization, and cell speed in HT1080 fibrosarcoma cells (*Sung et al., 2015*; *Sung et al., 2011*; *Sinha et al., 2016*), fibronectin was unable to rescue the filopodia defect of endoglin-KD HT1080 cells in this study (Fig. S7D). Instead, THSD7A was identified as a distinct exosome cargo that specifically promotes de novo filopodia formation in HT1080 and other cell types and rescued defects of exosome-inhibited cells in filopodia formation. Thus, multiple exosome cargoes contribute to diverse aspects of cytoskeletal reorganization and cell migration.

While no study has previously shown that EVs can induce filopodia formation, filopodia have previously been shown to interact with EVs in diverse ways. EVs were shown to bind and 'surf' on filopodia before entering cells (*Heusermann et al., 2016*). EVs have also been shown to bud from the surface of filopodia and other similar protrusions (*Dixson et al., 2023*; *McConnell et al., 2009*; *Nishimura et al., 2021*; *Rilla et al., 2013*; *Wang et al., 2021*). These various interactions influence cells in a variety of ways, including the delivery and release of EV cargoes and the promotion of cellular migration and invasion.

Filopodia are important for directional sensing, directional migration, and cell-cell contact formation in a variety of cell types (*Blake and Gallop, 2023*; *Arjonen et al., 2011*; *Jacquemet et al., 2015*). In cancer, filopodia formation is associated with aggressive behavior and cancer metastasis. Indeed, we found that endoglin-KD cancer cells had defects in not only filopodia formation but also in 3D migration and metastasis to chick embryo chorioallantoic membranes. In neurons, inhibition of exosome secretion reduced both filopodia and synapse formation. Thus, in two very different cell types, exosomes regulate these fundamental, filopodia-dependent behaviors. Future studies could further test the importance of exosomes and exosome-associated THSD7A in controlling filopodia in additional cell types, including endothelial cells that use filopodia during cell-cell contact formation and angiogenesis (*Fischer et al., 2019*). It would also be interesting to see whether THSD7A carried by EVs regulates kidney podocyte foot process formation downstream of filopodia and whether it is soluble proteolyzed THSD7A or THSD7A-carrying EVs released into the circulation by cancer cells that leads to autoimmune secondary membranous nephropathy (*Herwig et al., 2019*; *Wang et al., 2019*).

In our first proteomics dataset identifying SEV-enriched cargoes, we identified endoglin as a candidate molecule to regulate filopodia formation. While we originally hypothesized that endoglin was regulating filopodia formation through one of its known binding partners, our tests of candidates did not yield any positive results. Instead, through our proteomic comparison of control and endoglin-KD SEVs, we identified THSD7A as a possible filopodia-inducing EV cargo regulated by endoglin. Indeed, recombinant purified THSD7A rapidly induced filopodia formation, rescuing filopodia defects of endoglin-KD cells within 15 min (Fig. S7E). As neurons do not express endoglin (*Matsubara et al., 2000*) but do express THSD7A, we expect that sorting of THSD7A to exosomes is controlled by additional molecules. In addition, we found for the first time that Cdc42 activity was required for THSD7A to promote filopodia formation. However, the intermediary proteins that link THSD7A to Cdc42 remain unknown. Future studies to identify THSD7A-binding partners will be important to understand both the mechanism by which THSD7A selectively induces filopodia formation via Cdc42 and how its trafficking is regulated in neurons and other cell types.

Endoglin is highly expressed in endothelial cells and regulates angiogenesis (*Nassiri et al., 2011*). Indeed, mutations in endoglin are a frequent cause of hereditary hemorrhagic telangiectasia – a disease in which abnormal vascular structures are formed in the skin, mucous membranes, and some organs (*Govani and Shovlin, 2009*). Endoglin expression is also upregulated in several cancers (*Pérez-Gómez et al., 2010*), including melanoma (*Pardali et al., 2011*), ovarian cancer (*Steg et al., 2012*), breast cancer (*Davidson et al., 2010*), and gastric cancer (*Miao et al., 2016*). In breast cancer, these elevated endoglin levels are correlated with reduced survival and invasive phenotype (*Davidson et al., 2010*; *Oxmann et al., 2008*). There is also blood vessel enrichment of endoglin in ovarian

tumors (*Szubert et al., 2016*) and head and neck squamous cell carcinoma (*de Oliveira et al., 2014*). In fact, anti-endoglin antibodies are being evaluated as potential anti-cancer therapy and inhibit angiogenesis and metastasis in pre-clinical models (*Bernabeu et al., 2023*; *González Muñoz et al., 2021*; *Paauwe et al., 2016*; *Uneda et al., 2009*). Given our new findings that endoglin is a key regulator of filopodia via regulating THSD7A trafficking to SEVs, an important future direction could be to test the contribution of THSD7A to endoglin-dependent phenotypes, including angiogenesis and cancer metastasis.

In summary, we found that exosomes promote filopodia formation in diverse cell types by carrying the transmembrane ECM molecule THSD7A. This appears to be a major mechanism for filopodia formation with broad importance for a variety of cellular processes, including cancer cell migration and neuronal synapse formation.

## Limitations of the study

The main findings of this study are (1) exosomes induce filopodia formation in both cancer cells and neurons and synapse formation in primary neurons; (2) endoglin and THSD7A are exosomal cargoes that promote filopodia formation; (3) endoglin regulates the levels of THSD7A in cancer cell SEVs and promotes cancer metastasis; (4) THSD7A is expressed by both cancer cells and neurons and directly induces filopodia formation, including in cells that cannot secrete exosomes due to genetic engineering. As a first step to identify the mechanism by which THSD7A induces filopodia formation, we tested whether the candidate molecule Cdc42 is required downstream of THSD7A. We used the inhibitor ML141 and a dominant active Cdc42 molecule to show respectively that induction of filopodia by THSD7A is inhibited with ML141 and that activation of Cdc42 can overcome the filopodia defects of endoglin-KD. One limitation of the study is that we did not use a biochemical method to show that exosomes can activate Cdc42 in cells. Activation of Cdc42 is typically quantified by assessing the ratio of the GDP-bound inactive form to the GTP-bound active form via methods such as bead pulldown assays and Western blots. We were unable to detect active Cdc42 in our cancer cell lysates, suggesting the need to greatly scale up and further optimize in order to achieve the required signal for this assay. Thus, while our data show promising results to support the role of Cdc42 in this process, further experimentation using orthogonal methods such as Cdc42 activity assays and testing Cdc42 knockdown would strengthen this finding and be an important future direction. Another future direction could be to test the role of additional proteins known to be involved in filopodia formation, such as actin-binding proteins like mDia or ENA/VASP proteins, or BAR domain proteins. A second limitation is that we have not worked out the full mechanism by which endoglin regulates THSD7A secretion in exosomes. In order to do so, we would need to determine definitively how THSD7A trafficking is altered by endoglin, i.e. alterations in secretion vs. formation of THSD7A-containing EVs (*Figure 9A*). More in-depth assessments of the cellular distribution of THSD7A and endoglin, along with trafficking assays, would aid this endeavor.

## Materials and methods

**Key resources table**

| Reagent type (species) or resource | Designation | Source or reference | Identifiers | Additional information |
|---|---|---|---|---|
| Antibody | anti-CD63 (Mouse monoclonal) | Abcam | ab8219 | IF 1:500 |
| Antibody | anti-mouse Endoglin (Rabbit polyclonal) | Cell Signaling | 3290 | WB 1:1000 |
| Antibody | anti-human Endoglin (Rabbit monoclonal) | Cell Signaling | 4335 | WB 1:1000 |
| Antibody | anti-CD63 (Rabbit polyclonal) | Abcam | ab68418 | WB 1:1000 |
| Antibody | anti-Beta actin (Rabbit monoclonal) | Cell Signaling | 4970 | WB 1:5000 |

*Continued on next page*

*Continued*

| Reagent type (species) or resource | Designation | Source or reference | Identifiers | Additional information |
|---|---|---|---|---|
| Antibody | anti-TSG101 (Rabbit polyclonal) | Abcam | ab30871 | WB 1:1000 |
| Antibody | anti-Flotillin-1 (Rabbit polyclonal) | Cell Signaling | 3253 | WB 1:1000 |
| Antibody | anti-TGFbeta1 (Rabbit polyclonal) | Abcam | ab92486 | WB 1:1000 |
| Antibody | anti-HSP70 (Mouse monoclonal) | Santa Cruz | sc-24 | WB 1:1000 |
| Antibody | anti-CD29 (beta1-integrin; Mouse monoclonal) | BD Biosciences | 610467 | WB 1:1000 |
| Antibody | anti-ALK1/ACVRL1 (Rabbit polyclonal) | Abcepta | AP7807a | WB 1:1000 |
| Antibody | anti-Rab27a (Rabbit monoclonal) | Cell Signaling | 69295 | WB 1:1000 |
| Antibody | anti-Hrs (M-79) (Rabbit polyclonal) | Santa Cruz | sc-30221 | WB 1:1000 |
| Antibody | rabbit anti-GM130 (Rabbit monoclonal) | ThermoFisher | MA5-35107 | WB 1:1000 |
| Antibody | anti-THSD7A (Rabbit polyclonal) | Sigma | HPA000923 | WB 1:1000 |
| Antibody | anti-Vinculin (Mouse monoclonal) | Sigma-Aldrich | V9131 | WB 1:1000 |
| Antibody | anti-THSD7A (Rabbit polyclonal) | Nicola Tomas, University Medical Center Hamburg-Eppendorf | | WB 1:1000 |
| Antibody | anti-pSmad1/5/9 (Rabbit monoclonal) | Cell Signaling | 13820 | WB 1:1000 |
| Antibody | anti-Smad1 (Rabbit monoclonal) | Cell Signaling | 6944 | WB 1:1000 |
| Antibody | anti-pSmad2 (Rabbit monoclonal) | Cell Signaling | 3108 | WB 1:1000 |
| Antibody | anti-Smad2 (Rabbit monoclonal) | Cell Signaling | 5339 | WB 1:1000 |
| Antibody | Anti- SV2 (Mouse monoclonal) | Developmental Studies Hybridoma Bank | | IF 1:1000 |
| Antibody | anti-Flotillin-1 (Mouse monoclonal) | BD Biosciences | 610820 | WB 1:1000 |
| Antibody | anti-Alix (Mouse monoclonal) | Cell Signaling | 2171 | WB 1:1000 |
| Antibody | anti-GM130 (Mouse monoclonal) | BD Biosciences | 610822 | WB 1:1000 |
| Cell line (*Mus musculus*) | B16F1 | ATCC | | mouse melanoma |
| Cell line (*Homo sapiens*) | HT1080 | ATCC | | human fibrosarcoma |
| Recombinant DNA reagent | mCherry-pTαS2 (plasmid) | Donna J. Webb, Vanderbilt University | | Expression of mCherry in neurons |
| Recombinant DNA reagent | GFP-pTαS2 (plasmid) | Donna J. Webb, Vanderbilt University | | Expression of GFP in neurons |

*Continued on next page*

*Continued*

| Reagent type (species) or resource | Designation | Source or reference | Identifiers | Additional information |
|---|---|---|---|---|
| Recombinant DNA reagent | mScarlet-pTαS2 (plasmid) | This paper | | Expression of mScarlet in neurons, described in Neuron methods: Reagents and constructs |
| Recombinant DNA reagent | GFP-Rab27b (plasmid) | Addgene | 89447 | Expression of GFP-tagged human Rab27B; a gift from Wendy Westbroek to Addgene |
| Recombinant DNA reagent | pCMV-Tag4-THSD7A-FLAG (plasmid) | Yung-Jen Chuang, National Tsing Hua University, Taiwan | | Expression of THSD7A-FLAG |
| Recombinant DNA reagent | pCMV-Tag4 (plasmid) | Yung-Jen Chuang, National Tsing Hua University, Taiwan | | Empty vector for C-terminal FLAG-tagged expression |
| Recombinant DNA reagent | pCytERM_mScarlet_N1 (plasmid) | Addgene | 85066 | Expression of mScarlet-tagged cytochrome p450 aa 1–29; A gift from Dorus Gadella to Addgene |
| Recombinant DNA reagent | pCMV-Tag4-THSD7A-mScarlet (plasmid) | This paper | | Expression of mScarlet-tagged THSD7A, described in Cancer cell methods: Plasmids and reagents |
| Recombinant DNA reagent | pLenti6/V5-DEST (plasmid) | Thermo Fisher | V49610 | Empty vector for V5-tagged expression |
| Recombinant DNA reagent | pLenti6-WT-Eng-V5 (plasmid) | This paper | | Expression of shRNA-resistant WT endoglin-V5, described in Cancer cell methods: Plasmids and reagents |
| Recombinant DNA reagent | pRK5-myc-Cdc42-Q61L (plasmid) | Addgene | 12974 | Expression of myc-tagged Cdc42 constitutively active; a gift from Gary Bokoch to Addgene |
| Recombinant DNA reagent | pHluorin_M153R-CD63-mScarlet (plasmid) | BH Sung et al., Nat Commun., 2020 | | Expression of dual color CD63 reporter of exosome secretion and uptake |
| Recombinant DNA reagent | tdTomato-fTractin (plasmid) | Robert S. Fischer, NIH | | Expression of tdTomato-tagged F-actin. Provided by Robert Fischer |
| Transfected construct (*M. musculus*) | shHrs-1 | Dharmacon p-Blockit | | 5'-GGAACGAACCCAAGTACAAGG-3' |
| Transfected construct (*M. musculus*) | shHrs-2 | Dharmacon p-Blockit | | 5'-GCATGAAGACGAACCACATGC-3' |
| Transfected construct (*M. musculus*) | shRab27a-1 | Dharmacon p-Blockit | | 5'-GTGCGATCAAATGGTCATGCC-3' |
| Transfected construct (*M. musculus*) | shRab27a-2 | Dharmacon p-Blockit | | 5'-CGTTCTTCAGAGATGCTATGC-3' |
| Transfected construct (*M. musculus*) | pBlockit-shLacZ | Dharmacon p-Blockit | | control shRNA |
| Transfected construct (*M. musculus*) | shEng-1 | Dharmacon TRC lentiviral shRNA (pLKO.1) | TRCN0000094354 | 5'-TTAGGCTTCTAAGCAGCATGG-3' |
| Transfected construct (*M. musculus*) | shEng-2 | Dharmacon TRC lentiviral shRNA (pLKO.1) | TRCN0000094355 | 5'-TATAGATGACAAACAGCAGGG-3' |
| Transfected construct (*M. musculus*) | shScr | Dharmacon TRC lentiviral shRNA (pLKO.1) | | control shRNA |

*Continued*

| Reagent type (species) or resource | Designation | Source or reference | Identifiers | Additional information |
|---|---|---|---|---|
| Transfected construct (*H. sapiens*) | shEng-1 | Dharmacon TRC lentiviral shRNA (pLKO.1) | TRCN0000083138 | 5'-ATCCAGGTTCAAATGACAGGG-3' |
| Transfected construct (*H. sapiens*) | shEng-2 | Dharmacon TRC lentiviral shRNA (pLKO.1) | TRCN0000083141 | 5'-ATCATACTTGCTGACACCTGC-3' |
| Transfected construct (*H. sapiens*) | shEng | Dharmacon TRC lentiviral shRNA (pLKO.1) | TRCN0000083139 | 5'-TAGTGGTATATGTCACCTCGC-3' |
| Transfected construct (*H. sapiens*) | shTHSD7A | Dharmacon SMARTvector Lentiviral shRNAs | V3SH11240-226004853 | N/A, proprietary to Dharmacon |
| Transfected construct (*H. sapiens*) | shTHSD7A | Dharmacon SMARTvector Lentiviral shRNAs | V3SH11240-226644723 | N/A, proprietary to Dharmacon |
| Transfected construct (*H. sapiens*) | shTHSD7A | Dharmacon SMARTvector Lentiviral shRNAs | V3SH11240-227287134 | N/A, proprietary to Dharmacon |
| Transfected construct (*H. sapiens*) | NTC (non-targeting control) | Dharmacon SMARTvector Lentiviral shRNAs | | N/A, proprietary to Dharmacon |
| Transfected construct (*Rattus norvegicus*) | Hrs shRNA | pSUPER, doi: 10.1126/science.1068999 | | 5'-AATGAACCCAAGTACAAGG-3' For transfection of neurons |
| Transfected construct (*R. norvegicus*) | Hrs shRNA | pSUPER Vector, doi: 10.1126/science.1068999 | | 5'-ATGAAGAGCAACCACATGA-3' For transfection of neurons |
| Transfected construct (*R. norvegicus*) | Rab27b shRNA | pSUPER Vector, doi: 10.1126/science.1068999 | | 5'-CCGATACACAGACAATAAA-3' For transfection of neurons |
| Transfected construct (*R. norvegicus*) | Rab27b shRNA | pSUPER Vector, doi: 10.1126/science.1068999 | | 5'-TGAGTCAACTGCAGGCAAA-3' For transfection of neurons |
| Transfected construct (*M. musculus*) | Endoglin siRNA | Dharmacon ON-TARGETplus siRNA SMARTpool | L-045109-00-0005 | N/A, proprietary to Dharmacon |
| Transfected construct (*M. musculus*) | NTC (non-targeting control) | Dharmacon | D-001810-01-05 | N/A, proprietary to Dharmacon |
| Peptide, recombinant protein | human BMP-9 | R&D Systems | 3209 BP | recombinant protein |
| Peptide, recombinant protein | human TGF-beta1 | R&D Systems | 240-B-002 | recombinant protein |
| Peptide, recombinant protein | human Fibronectin | Cultrex | 3420-001-01 | recombinant protein |
| Peptide, recombinant protein | human THSD7A | R&D Systems | 9524-TH | recombinant protein |
| Other | EZSHPERE 96-well plate | SGC Techno Glass | AG4860900SP | spheroid collagen invasion assay plate |

*Continued on next page*

*Continued*

| Reagent type (species) or resource | Designation | Source or reference | Identifiers | Additional information |
|---|---|---|---|---|
| Other | type 1 A collagen solution | Nitta gelatin | Type-1-A | spheroid collagen invasion culture medium |
| Chemical compound, drug | ML141 | Millipore | 217708 | Cdc42 inhibitor |
| Other | OptiPrep Density Gradient Medium | Sigma | D1556 | iodixanol-based solution for density gradient |
| Commercial assay, kit | Coomassie Brilliant Blue R-250 Staining Solutions Kit | BioRad | 1610435 | protein stain |
| Chemical compound, drug | Phenylmethylsulfonyl fluoride (PMSF) | Research Products International | P20270 | Serine protease inhibitor |
| Chemical compound, drug | cOmplete Protease Inhibitor Cocktail | Roche | 4693116001 | protease inhibitor cocktail |
| Chemical compound, drug | TGF-beta1 inhibitor SB 431542 | Sigma-Aldrich | S4317 | |
| Chemical compound, drug | PhosSTOP | Roche | 4906845001 | Phosphatase inhibitor cocktail |
| Commercial assay or kit | BCA Protein Assay Kit | Pierce | 23225 | Protein quantification assay |
| Commercial assay or kit | Micro BCA Protein Assay Kit | Pierce | 23235 | Protein quantification assay |
| Chemical compound, drug | Lipofectamine RNAiMAX | Life Technologies | 13778150 | transfection reagent |
| Chemical compound, drug | Lipofectamine 2000 | Invitrogen | 11668–027 | transfection reagent |

## Materials availability statement

The pLenti6-WT-Eng-V5 and pCMV-Tag4-THSD7A-mScarlet plasmids were newly created for these experiments and can be requested by contacting Alissa Weaver.

## Cancer cell methods

### Cell culture

Cell lines were obtained from ATCC and checked regularly for mycoplasma contamination. Cells used for experiments were mycoplasma-negative. B16F1 mouse melanoma cells (ATCC CRL-6323) were maintained in DMEM supplemented with 10% FBS. HT1080 human fibrosarcoma cells (ATCC CCL-121) were maintained in DMEM supplemented with 10% bovine growth serum (BGS). Cells were maintained at 37°C in 5% $CO_2$. Transient transfections were done using Lipofectamine 2000 (Invitrogen #11668–027).

### Plasmids and reagents

Stable knockdown of Hrs in B16F1 cells was achieved by using the Dharmacon p-Blockit system with target sequences: 5'-GGAACGAACCCAAGTACAAGG-3' (shHrs-1) and 5'-GCATGAAGACGA ACCACATGC-3' (shHrs-2), stably selected with blasticidin. Stable knockdown of Rab27a in B16F1 cells was achieved by using the p-Blockit system with target sequences: 5'-GTGCGATCAAATGGTC ATGCC-3' (shRab27a-1) and 5'-CGTTCTTCAGAGATGCTATGC-3' (shRab27a-2), stably selected with blasticidin. Control lines were simultaneously selected using the pBlockit-shLacZ plasmid. Stable knockdown of endoglin in B16F1 cells was achieved using the Dharmacon TRC lentiviral shRNA (pLKO.1): 3'UTR target shEng-1 (TRCN0000094354): 5'-TTAGGCTTCTAAGCAGCATGG-3', ORF target shEng-2 (TRCN0000094355): 5'-TATAGATGACAAACAGCAGGG-3'. B16F1 shEng cells were stably selected using puromycin. The control line was simultaneously selected using the shScr control plasmid. Transient endoglin knockdown in B16F1 cells was obtained with ON-TARGETplus siRNA SMARTpool (L-045109-00-0005; GE Dharmacon) using Lipofectamine RNAiMAX (Life Technologies).

As a control for the knockdown, a nontargeting control pool was used (D-001810-01-05; GE Dharmacon). Stable knockdown of endoglin in HT1080 (mCitrine expressing) cells was achieved using Dharmacon TRC lentiviral shRNA (pLKO.1): 3'UTR shEng-1 (TRCN0000083138): 5'-ATCCAGGTTCAAATGACAGGG-3', ORF shEng-2 (TRCN0000083141): 5'-ATCATACTTGCTGACACCTGC-3', CDS shEng (TRCN0000083139): 5'-TAGTGGTATATGTCACCTCGC-3'. Cells were selected with puromycin and maintained in blasticidin and puromycin to retain mCitrine expression and endoglin knockdown. The control line was simultaneously selected using the shScr control plasmid. Knockdown of THSD7A in HT1080 cells was achieved using Dharmacon SMARTvector Lentiviral shRNAs Cat# V3SH11240-226004853, V3SH11240-226644723, and V3SH11240-227287134. A non-targeting control (NTC) shRNA was used as a control.

The pCMV-Tag4-THSD7A-FLAG plasmid and vector control pCMV-Tag4 were generously provided by the Chuang laboratory (*Kuo et al., 2011*). The pCMV-Tag4-THSD7A-mScarlet plasmid was created by subcloning the mScarlet region from pCytERM_mScarlet_N1 (Addgene #85066) into the C-terminal end of THSD7A in the pCMV-Tag4-THSD7A plasmid. For mechanistic studies, the END gene was amplified from HT1080-derived cDNA and subcloned into the pENTR vector. Silent mutations were inserted to confer shEng resistance, and the coding sequence was confirmed by Sanger sequencing. The resulting pENTR-Eng plasmid was Gateway recombined with pLenti6/V5-DEST, in which the *BSD* (blasticidin resistance) gene in pLenti6/V5-DEST was replaced with *neo* to confer neomycin resistance. HT1080 shEng cells were transfected with pLenti6/V5-DEST or pLenti6-WT-Eng-V5 and stable lines were isolated by selection with G418 sulfate. pRK5-myc-Cdc42-Q61L was purchased from Addgene (Plasmid #12974). HT1080 cells stably expressing mCitrine fluorescent protein were obtained from Andries Zijlstra's laboratory. HT1080 cells stably expressing the dual tagged CD63 live imaging reporter pHluorin_M153R-CD63-mScarlet were previously described (*Sung and Weaver, 2023*).

## EV isolation and characterization

48 hr conditioned Opti-MEM (a serum-free but growth factor-containing medium) was collected for isolation of EVs from cancer cells. A 5-min 300 x *g* spin and 25-min 2000 x *g* spin were performed to remove dead cells and large particulate matter. The supernatant was then spun at 10,000 x *g* for 30 minutes to pellet the LEVs (Ti45 rotor, Beckman Coulter). This LEV fraction was later washed with PBS spins and used for western blotting and rescue experiments. To isolate SEVs from cancer cells, we used a cushion density gradient ultracentrifugation method, which reduces vesicle and protein aggregation and leads to a highly purified preparation (*Li et al., 2018*). For iodixanol cushion and gradient, OptiPrep Density Gradient Medium was purchased from Sigma (D1556). The supernatant from the 10,000 x *g* spin was layered on top of a 2 ml 60% iodixanol cushion and spun for 4 hr at 100,000 x *g* (SW32 Ti rotor, Beckman Coulter). The bottom 3 ml of the cushion was obtained and layered at the bottom of a discontinuous gradient, followed by three 3 ml layers of 20%, 10%, and 5% iodixanol diluted with 0.25 M sucrose/10 mM Tris, pH 7.5. The discontinuous gradient was spun for 18 hours at 100,000 x *g* and collected into 12 1-ml fractions (SW40 Ti rotor, Beckman Coulter). SEVs are typically located in fractions 6 and 7, and these fractions are then washed by resuspending with PBS and repelleting. After final resuspension with PBS, SEV purification is checked using nanoparticle tracking for particle number and size, western blot validation of EV proteins, and electron microscopy for morphology (*Lötvall et al., 2014*; *Théry et al., 2018*; *Welsh et al., 2024*). Large volumes of conditioned media were concentrated prior to the cushion centrifugation step using Vivacell 70 ultrafiltration units with 100,000 MWCO. The number of vesicles added back in rescue experiments was determined by calculating the physiological SEV secretion rate for each cell line. The total vesicle number was counted using nanoparticle tracking analysis (Particle Metrix ZetaView or NanoSight), and the vesicle per cell per hour secretion rate was calculated. This rate was used to determine a range of physiologically relevant vesicle numbers to treat cells.

## Electron microscopy

Freshly isolated SEVs or LEVs suspended in PBS were used for TEM imaging. Samples were incubated on glow-discharged 300 mesh carbon-coated grids for 30 s followed by fixation in 1% glutaraldehyde for 1 min. Samples were washed twice and negative stained in 2% uranyl acetate. TEM was performed on a Tecnai T12 operating at 100 keV using an AMT nanosprint5 CMOS camera.

## Immunofluorescence and analysis

Cells were seeded onto 100 µg/ml PDL-coated glass coverslips. For all experiments, fields of view containing stained cells were randomly selected for imaging and filopodia quantification. For steady-state filopodia quantification, cells were fixed with 4% paraformaldehyde, permeabilized with 0.1% Triton X-100 or 0.1% saponin, and stained with Alexa Fluor conjugated phalloidin (rhodamine-phalloidin or phalloidin-Alexa Fluor 488) 24 hr after seeding. Co-staining was done using mouse anti-CD63 (Abcam ab8219) and goat anti-mouse Alexa Fluor 546. For EV rescue experiments, SEVs or LEVs are added to live cells 24 hr post-seeding, and cells are fixed and stained with conjugated phalloidin 18 hr post-EV treatment. For BMP-9 treatment, human recombinant BMP-9 (R&D Systems Cat# 3209 BP) was reconstituted in sterile 4 mM HCl +0.1% BSA, diluted in OptiMEM, and added to cells for 1 hr at 37°C, then cells are fixed and stained with conjugated phalloidin. For TGF-β1 treatment, desired concentration of recombinant human TGF-β1 (R&D Systems Cat #240-B-002; reconstituted in sterile 4 mM HCl +0.1% BSA) was diluted in OptiMEM, added to cells at time of seeding, and cells were fixed and stained 48 hr post-seeding. For FN coating experiments, glass coverslips were coated with indicated concentrations of FN (Cultrex Human Fibronectin Cat# 3420-001-01, diluted in sterile PBS) overnight and cells were seeded onto coverslips. Cells were fixed and stained 24 hr post-seeding. For THSD7A rescue experiments, coverslips were first coated with 100 µg/ml PDL overnight, then washed, then coated with indicated concentrations of recombinant human THSD7A overnight (diluted in sterile PBS; R&D Systems Cat# 9524-TH). Cells were seeded onto THSD7A-coated coverslips and fixed and stained 24 hr post-seeding. For THSD7A-mScarlet expression experiments, THSD7A-mScarlet was transiently transfected into HT1080 cells and cells that were visibly expressing the fusion protein were imaged using a Nikon A1R HD25 confocal microscope with a Plan Apo $\lambda$ 60 x/1.4 oil objective. Cells were seeded onto PDL-coated glass coverslips overnight, then fixed and stained for phalloidin and CD63. Cell edges were manually drawn and only the internal THSD7A-mScarlet and CD63 signals were quantified. JACoP Fiji plugin was used to calculate M1 value from manually thresholded channels. All fixed cells were imaged using either a Nikon Eclipse TE2000-E epifluorescence microscope and Metamorph software (Molecular Devices) or Nikon A1R confocal microscope and NIS-Elements Software. Cell area was calculated by tracing cell outlines using ImageJ software. Filopodia were counted using the Filoquant ImageJ plugin *Jacquemet et al., 2017* with manual adjustment for any plugin errors.

## Live cell imaging

### MVE fusion movies

HT1080 cells stably expressing pHluorin_M153R-CD63-mScarlet were cultured on PDL-coated glass-bottomed MatTek dishes. Cells were imaged every 10 s for 20 min on a Nikon A1R confocal microscope using a Plan Apo $\lambda$ 60x/1.4 oil objective and a humidified 37°C chamber with 5% $CO_2$. MVE fusion events are identified as MVEs expressing mScarlet-CD63 approaching the cell edge followed by a burst of green fluorescence, suggesting exposure to extracellular neutral pH and MVE fusion. Filopodia that arose at the region of MVE fusion soon after were counted in the quantification. CD63 green burst and filopodia formation were noted based on the time frame, and the timing between these events was quantified and graphed.

### Filopodia dynamics

B16F1 control (shLacZ) and Hrs-KD cell lines were transiently transfected with tdTomato-fTractin 2 days prior to seeding onto PDL-coated glass-bottomed MatTek dishes. Prior to imaging, cell media was switched to Leibovitz's L-15 (Gibco)+10% FBS. Cells were imaged on Nikon Eclipse TE2000-E epifluorescence microscope using Apo 60x oil objective and warmed 37°C chamber and an image was captured every 30 s for 15 min with MetaMorph software (Molecular Devices). B16F1 control (shScr) and Eng-KD cell lines were transiently transfected with tdTomato-fTractin 2 days prior to seeding onto PDL-coated glass-bottom MatTek plates. Cells were imaged on Nikon A1R confocal microscope using Plan Apo $\lambda$ 60 x/1.4 oil objective and a humidified 37°C chamber with 5% $CO_2$. An image was captured every 30 s for 15 min with NIS-Elements software. Only filopodia that were formed and completely retracted during the 15-min time period were quantified as de novo, and lifetime was defined as the total amount of time the filopodia persisted from formation to complete retraction.

## Proteomics analyses

### Identification of individual bands isolated from colloidal Coomassie blue-stained gel

Purified LEVs and SEVs isolated from B16F1 cells were run on a standard SDS-PAGE gel using fresh buffers and after washing the gel apparatus and cleaning it with 70% ethanol. After staining and destaining with the BioRad Coomassie Brilliant Blue R-250 Staining Solutions Kit (#1610435), four bands that were apparent in the SEV sample but missing in the LEV sample were identified, cut from the gel, and submitted for proteomics analysis using trypsin as a digestion enzyme. MS/MS samples were analyzed using Sequest (Thermo Fisher Scientific, San Jose, CA, USA; version 27, rev. 12) and X! Tandem (The GPM, thegpm.org; version CYCLONE (2010.12.01.1)). Sequest was set up to search the uniprot-mouse-reference-canonical_20121112_rev database (unknown version, 86222 entries), assuming the digestion enzyme trypsin. X! Tandem was set up to search the uniprot-mouse-reference-canonical_20121112_rev database (unknown version, 86222 entries), also assuming trypsin. Sequest was searched with a fragment ion mass tolerance of 0.00 Da and a parent ion tolerance of 2.5 Da. X! Tandem was searched with a fragment ion mass tolerance of 0.50 Da and a parent ion tolerance of 2.5 Da. Glu->pyro->Glu of the n-terminus, ammonia-loss of the n-terminus, gln->pyro->Glu of the N-terminus, oxidation of methionine and carbamidomethyl of cysteine were specified in X! Tandem as variable modifications. Oxidation of methionine and carbamidomethyl of cysteine was specified in Sequest as variable modifications. Criteria for protein identification: Scaffold (version Scaffold_4.8.8, Proteome Software Inc, Portland, OR) was used to validate MS/MS based peptide and protein identifications. Peptide identifications were accepted if they could be established at greater than 50.0% probability to achieve an FDR less than 5.0% by the Peptide Prophet algorithm (Keller, A. et al. Anal. Chem. 2002;74(20):5483–92). Protein identifications were accepted if they could be established at greater than 81.0% probability to achieve an FDR less than 5.0% and contained at least two identified peptides. Protein probabilities were assigned by the Protein Prophet algorithm (Nesvizhskii, Al et al. Anal. Chem. 2003;75(17):4646–58). Proteins that contained similar peptides and could not be differentiated based on MS/MS analysis alone were grouped to satisfy the principles of parsimony. The full proteomics results are shown in *Supplementary file 1*.

### iTRAQ analysis of SEVs

For iTRAQ proteomics analysis, isolated exosomes from multiple cushion-density gradient preparations were pooled together for each cell type. Each individual preparation was tested for purity using Zetaview nanoparticle tracking and Western blot for typical exosomal markers. Pooled SEVs were resuspended in PBS and mixed 1:1 with 2x 'exosome lysis buffer' (200 mM TEAB, 600 mM NaCl, 2% NP-40, and 1% sodium deoxycholate). After mixing, samples were then sonicated using a Bioruptor in cold ice water (30 s on/30 s off for 15 min). After sonication, samples were spun down to pellet insoluble proteins, and the supernatant was submitted and run by the Vanderbilt University MSRC Proteomics Core Laboratory. MS/MS spectra were searched against a mouse subset of the UniProt KB protein database, and autovalidation procedures in Spectrum Mill were used to filter the data to <1% false discovery rates at the protein and peptide level. The median log2 iTRAQ protein ratios were calculated over all peptides identified for each protein, and frequency distributions were generated in GraphPad Prism. Log2 ratios typically follow a normal distribution and were fit using least squares regression. The mean and standard deviation values derived from the Gaussian fit were used to calculate p-values using Z-score statistics. A given iTRAQ protein ratio, with the calculated mean and standard deviation of the fitted dataset, is transformed to a standard normal variable ($z = (x-\mu)/\sigma$). Since the properties of the standard normal curve are known, area under the curve for a particular value can be calculated, providing a p-value for each measured protein ratio. Calculated p-values were subsequently corrected for multiple comparisons using the Benjamini-Hochberg method. The full proteomics results are shown in *Supplementary file 2*.

### Western blot analysis

Samples for Western blotting were run on a reducing SDS-PAGE gel and transferred to a nitrocellulose membrane (unless otherwise noted). Cell lysate samples were collected by scraping cells directly from a tissue culture dish using RIPA cell lysis buffer (50 mM Tris pH 7.6, 150 mM NaCl, 1% NP-40, 1%

SDS, 0.5% sodium deoxycholate) with 1 mM Phenylmethylsulfonyl fluoride (PMSF) (Research Products International Cat# P20270) and cOmplete Protease Inhibitor Cocktail (Roche Cat# 04693116001, used as directed by manufacturer). EV samples were lysed by mixing directly with Laemmli sample buffer containing SDS, DTT, and BME. Samples were loaded at equal protein amounts; protein amount was quantified from samples by using a BCA assay or micro BCA assay (Pierce BCA Protein Assay Kit, Thermo Fisher Scientific Cat# 23225; Micro BCA Protein Assay Kit, Thermo Fisher Scientific Cat# 23235). Antibodies (for cancer cells): rabbit anti-Endoglin (mouse specific) (Cell Signaling Cat# 3290), rabbit anti-Endoglin (human specific) (Cell Signaling Cat# 4335), rabbit anti-CD63 (Abcam Cat# ab68418), rabbit anti-beta actin (Cell Signaling Cat# 4970), rabbit anti-TSG101 (Abcam Cat# ab30871), rabbit anti-flotillin-1 (Cell Signaling Cat# 3253), rabbit anti-TGFbeta1 (Abcam Cat# ab92486), mouse anti-HSP70 (Santa Cruz Cat# sc-24), mouse anti-CD29 (beta1-integrin; BD Biosciences Cat# 610467), rabbit anti-ALK1/ACVRL1 (Abcepta Cat# AP7807a), rabbit anti-Rab27a (Cell Signaling Cat# 69295), rabbit anti-Hrs (M-79) (Santa Cruz Cat# sc-30221), rabbit anti-GM130 (Thermo Fisher Cat# MA5-35107), rabbit anti-THSD7A (Sigma Cat# HPA000923), mouse anti-vinculin (Sigma-Aldrich Cat# V9131).

To detect THSD7A in HT1080 TCLs and SEVs, reducing western blot conditions and rabbit anti-THSD7A (Sigma HPA000923) were used. To detect THSD7A in B16F1 SEV samples, SEVs were solubilized in NP-40 lysis buffer (50 mM Tris pH 8.0, 150 mM NaCl, 1.0% NP-40, 1 mM PMSF mixed 1:1 with SEVs resuspended in PBS) in non-reducing conditions and ran on native gels using non-SDS running buffer. After transfer, the membrane was probed with rabbit anti-THSD7A IgG, generously provided by Nicola M. Tomas, Rolf A.K. Stahl, and Friedrich Koch-Nolte at University Medical Center Hamburg-Eppendorf in Hamburg, Germany (*Tomas et al., 2017*).

For detecting Smad phosphorylation, TGF-β1 was used as a positive control and added to indicated conditions for 1 hr at 10 ng/ml. TGF-β1 was reconstituted in sterile 4 mM HCl +0.1% BSA according to manufacturer's instructions (R&D Systems Cat# 240-B-002). Indicated conditions were treated with the TGF-β1 inhibitor SB 431542 (Sigma-Aldrich Cat# S4317; solubilized in DMSO) for 5 min at 10 µM prior to TGF-β1 treatment. Cells were seeded on 100 µg/ml PDL +/-2 µg/ml rhTHSD7A. At experiment endpoint, media was aspirated and adherent cells were lysed with RIPA lysis buffer (50 mM Tris pH 7.6, 150 mM NaCl, 1% NP-40, 1% SDS, 0.5% sodium deoxycholate) containing 1 mM phenylmethylsulfonyl fluoride (PMSF) (Research Products International (RPI) Cat# P202705) and PhosSTOP (Roche Cat# 4906845001). Lysates were passed through a 27-gauge needle three times prior to boiling and preparing for Western blot. pSmad1/5/9 (Cell Signaling #13820) and Smad1 (Cell Signaling #6944) were probed on one membrane and pSmad2 (Cell Signaling #3108) and Smad2 (Cell Signaling #5339) were probed on a separate membrane. Membranes were blocked with 5% BSA in TBS/0.1% Tween-20 overnight, probed with indicated phospho-antibody and then stripped and reprobed with indicated total Smad antibody.

## Avian embryo model of metastasis

Avian embryo experimental metastasis model protocol was based on previously published methods (*Palmer et al., 2011*). At the time points of the experiments, the chicken embryos are not considered vertebrate animals and therefore are not subject to IACUC regulation. Live fertilized chicken eggs were incubated for 11 days prior to injections. Eggs were chosen for injection based on the ease of visualizing the allantoic vein. Eggs with clear vein visibility were chosen at random for cell and SEV injection conditions. HT1080 control and KD cells expressing mCitrine were suspended in PBS at a concentration of 1 x 10⁶ cells/mL and 100,000 cells (100 µl) were injected into the allantoic vein in the direction of blood flow. Eggs were returned to the incubator for 4 days. After 4 days post-injection, chicks were sacrificed, and three circular areas of CAM were harvested from inside the shell 4 days post-injection. The membrane was then peeled away from the eggshell and placed between a glass slide and coverslip for imaging. 25–30 fields of view were captured for each egg harvested and 4–7 eggs were sacrificed for each condition. Sample size was determined by time constraints of number of embryos that could be sacrificed and immediately imaged within a 12-hr time period. Embryos that died during the prior 4-day incubation period were excluded from analysis. Preliminary low-power wide-field images were obtained using a Zeiss Lumar V12 fluorescence stereomicroscope with ×10 magnification. For higher power wide-field images, fluorescent cell extravasation and metastasis were observed using X10/0.40 UPlanSApo objective lens with a 10X ocular (100X total) on an Olympus

BX-61 microscope equipped with a digital camera controlled with Volocity image acquisition software. For SEV rescue experiments, control SEVs were isolated using the cushion-density gradient method and were premixed with cells immediately prior to injection into chick embryo intravascularly. The number of SEVs per injection was calculated using the SEV secretion rate for the HT1080 shScr cell line according to data collected with the Zetaview nanoparticle tracking instrument ($89 \times 10^6$ SEVs per 100,000 cells). Images were analyzed in ImageJ by manually thresholding each image to include visible cells and avoid capturing background signals. The colony number and colony area of each thresholded image were quantified.

## Spheroid collagen invasion model

$1.5 \times 10^4$ HT1080 cells were cultured in an EZSPHERE micro-fabricated 96-well plate (AGC Techno Glass, Shizuoka, Japan). After 20 hr, the spheroids from each well were collected into a 15 ml tube by adding 5 ml of DMEM medium and centrifuging at $300 \times g$ for 1 min. For collagen gel culture, a cold 3 mg/ml type 1A collagen solution (Nitta gelatin, Osaka, Japan Cat# Type-1-A) was neutralized by adding an 8:1:1 ratio of collagen: 10 x DMEM/F12: reconstitution buffer 200 mM HEPES, 50 mM NaOH, and 260 mM NaHCO3, then diluted in cold PBS to achieve a final concentration of 1.2 mg/ml. 10 μl of the neutralized collagen/medium solution was coated onto a 35 mm glass bottomed culture dish and solidified at 37°C for 30 min. The collected spheroids were suspended in 100 μl of the ice-cold collagen/medium solution, domed onto the lower collagen gel, and solidified at 37°C for 30 min. 2 ml of the complete medium was gently added to the dish, and Time-Lapse images were captured using a Keyence digital fluorescence microscope BZ-9000 using the 10X phase contrast objective lens.

## Cdc42 inhibition assay

Cells were transiently transfected with constitutively active Cdc42-Q61L 24 hr prior to seeding on 100 μg/ml PDL coated coverslips with or without 2 μg/ml rhTHSD7A. 24 hr after seeding, cells were treated with either vehicle (DMSO) or 10 μM ML141 (Millipore Cat# 217708) diluted in OptiMEM media for 60 min. After treatment, cells were fixed with 4% paraformaldehyde and stained with phalloidin to visualize filopodia and Hoechst to visualize nuclei.

## **Neuron methods**

### Reagents and constructs

mCherry-pTαS2 and GFP-pTαS2 vectors were a kind gift from the laboratory of Donna Webb at Vanderbilt University (*Shi et al., 2013*). Stable knockdown in neurons was achieved by using the pSuper system with target sequences against Hrs: 5'-AATGAACCCAAGTACAAGG-3' and 5'- ATGA AGAGCAACCACATGA-3', and against Rab27b: 5'- CCGATACACAGACAATAAA-3' (shRab27b-1) and 5'-TGAGTCAACTGCAGGCAAA-3'. SV2 monoclonal antibodies were obtained from the Developmental Studies Hybridoma Bank (University of Iowa, Iowa City, IA). Alexa Fluor 488 Anti-Rabbit and Alexa Fluor 647 Anti-Mouse were from Molecular Probes. For neuronal cultures, B27 media was prepared by adding 2% B27 supplement and L-glutamine to neurobasal media.

### Primary cultures of neurons

Rat hippocampal and cortical neurons were isolated from day 19 embryos and plated on 1 mg/ml Poly-L-Lysine (PLL) or 50 μg/ml Poly-D-Lysine coated glass coverslips. Low-density cultures were prepared as described previously. In brief, the hippocampus and cortex were removed from dissected brains of day 19 rat embryos and incubated in 0.05% trypsin in HBSS for 10 min at 37°C. Neurons were washed with HBSS, homogenized by gentle mixing, and plated on PLL or PDL-coated coverslips. After 3–4 hr, neuron coverslips were transferred to 60 mm dishes containing primary astroglia for co-culture to promote neuronal health (*Kaech and Banker, 2006*).

### SEV isolation and characterization

For neuronal SEVs, day in vitro (*DIV*) 9 cortical neurons cultured at high density (2.6 million in 100 mm culture dishes) were washed three times with HBSS. After the final wash, HBSS was replaced with 4 ml Neurobasal media per 100 mm dish. Neurobasal media does not contain serum but contains growth factors. Neurobasal conditioned media was collected after 4 hr incubation and processed

for differential ultracentrifugation. Briefly, conditioned media was centrifuged sequentially at 300 x *g* for 10 min, 2000 x *g* for 25 min in a tabletop centrifuge, and 10,000 x *g* for 30 min in a Type 45 Ti ultracentrifuge rotor (Beckman) to remove live cells, cell debris, and LEVs, respectively. The supernatant from the 10,000 x *g* spin was centrifuged at 100,000 x *g* for 18 hr in a Type 45 Ti rotor to obtain SEVs. The 100,000 x *g* SEV-containing pellets were resuspended in 3 ml sterile cold PBS and repelleted at 100,000 x *g* for 4 hrs in a TLA110 rotor (Beckman). SEVs were analyzed for size and number by nanoparticle tracking (ZetaView, ParticleMetrix), for morphology by TEM, and for common SEV markers by western blotting.

## Western blot analysis

Neuron TCL and EV samples for western blot were prepared as described above in the cancer cell Western blot methods section. Antibodies (for neurons): rabbit anti-TSG101 (Abcam Cat# ab30871), mouse anti-Flotillin-1 (BD Biosciences Cat# 610820), Mouse anti-Alix (Cell Signaling Cat# 2171), mouse anti-GM130 (BD Biosciences Cat# 610822), rabbit anti-THSD7A (Sigma Cat# HPA000923).

## Calcium phosphate transfection

Neurons plated at low density on coated glass coverslips were transfected with a modified calcium phosphate transfection method at day 3 or 5 as previously described (*Lin et al., 2013*). 1 µg of mCherry-pTαS2 and either 1 µg of GFP-pTαS2 or 3 µgs of GFP-Rab27b/ shRNAs or 2 µgs of pCMV-FLAG/pCMV-FLAG-THSD7A were mixed with 120 µl of 250 mM $CaCl_2$ in an Eppendorf tube. 120 µl of 2X HEPES Buffered Saline (HBS; 274 mM NaCl, 9.5 mM KCl, 15 mM glucose, 42 mM HEPES, 1.4 mM $Na_2HPO_4$, pH 7.15) was added drop by drop to the DNA-$CaCl_2$ mixture with continuous aeration and incubated at room temperature for 15 min. The neuron coverslips were removed from co-cultures and transferred to sterile petri dishes, containing glial conditioned media, for dropwise addition of the transfection mixtures. Neurons were kept in the incubator at 37°C for 30–40 min and then washed three times with HBSS (135 mM NaCl, 4 mM KCl, 2 mM $CaCl_2$, 1 mM $MgCl_2$, 10 mM glucose, 20 mM HEPES, pH 7.35). The neuron coverslips were then transferred back to the home dishes containing astrocytes. The transfection efficiency was in the range of 5–10% using this method. Knockdown or expression of target genes was analyzed by immunofluorescence together with expression of co-transfected mCherry filler (shRNAs, see *Figure 3—figure supplement 2*) or fluorescence (GFP or GFP-Rab27b). Analysis of KD phenotypes was performed in transfected neurons, as assessed by fluorescence of the mCherry filler, as in previous publications (*Evans et al., 2015*).

## Immunocytochemistry

For most antibodies, neurons were fixed in 4% paraformaldehyde in PBS for 15 min at RT and then permeabilized with 0.2% triton-X in PBS for 5 min at RT. For PSD95 staining, neurons were fixed in 4% paraformaldehyde in PBS for 3 min at RT and then incubated in cold methanol for 10 min at –20°C. Next, neurons were incubated for 1 hr with 20% goat serum to block non-specific antibody binding. Primary antibody was diluted in 5% goat serum in PBS with 1:1000 dilutions of antibody. Neurons were incubated with the primary antibody mixture overnight at 4°C. The secondary antibody mixture was prepared similarly to the primary antibody mixture. Neurons were incubated for 1 hr at room temperature with the secondary antibody mixture. Cells were washed three times with 1x PBS between each step. After the final wash with PBS, neuron coverslips were mounted on glass slides using AquaPoly Mount and allowed to dry overnight.

## Microscopy and image analysis

Neuronal imaging was performed on a Quorum Wave-FX Yokogawa CSU-X1 spinning disk confocal system with a Nikon Eclipse Ti microscope. Images were acquired using MetaMorph software (Molecular Devices, Sunnyvale, CA) and a Plan Apo TIRF 60x (NA 1.49) objective. Images for GFP, mCherry, and SV2 647/PSD95 647 were acquired by laser excitation at 491 nm, 561 nm, and 642 nm, respectively. Emission filters for these fluorophores were 525/50, 593/40, or 620/60 and 700/75, respectively (Semrock, Rochester, NY). Primary or secondary dendrites from confocal images were randomly selected for quantification of filopodia and spine density. Dendritic filopodia were defined as thin headless protrusions. Dendritic spines were identified as dendritic protrusions that co-localize with

synaptic markers SV2. Synapses were defined as SV2-positive puncta present on both dendritic protrusions and dendritic shafts.

## Statistical analysis

The figure legends list the n values and independent experiment numbers for both quantitated data and representative images from experiments. All datasets were tested for normality using GraphPad Prism's Kolmogorov-Smirnov or Shapiro-Wilk normality test. No sample or data points were omitted from analysis. The two-sided unpaired Mann-Whitney or two-sided paired Wilcoxon test compared nonparametric data groups. Parametric data were compared using a two-sided unpaired or paired Student's t-test. Scatter plots were plotted with median and interquartile range or mean ± standard error of the mean. Bar graphs were plotted as mean ± standard error of the mean. All graphs were created using GraphPad Prism.

## Adherence to community standards

Characterization of extracellular vesicle populations was done in accordance with the 'Minimal information for studies of extracellular vesicles' (MISEV) published guidelines (*Théry et al., 2018*; *Welsh et al., 2024*).

## Acknowledgements

Funding was provided by NIH R01GM117916, R01CA206458, R01CA249684, and R01CA249424 grants to AMW, by AHA fellowship 17POST33660473 and NIH training grant T32CA009592 support of COM, by Grant-in-Aid for JSPS KAKENHI (JP 17K15005) to DH, by NIH R50CA283661 to BHS. This project was begun as a collaborative project with Dr. Donna Webb, who passed away before her time. We acknowledge her important scientific contributions to this work, including her mentorship of Dr. Mikin Patel, her conceptual contributions to the project, expertise in neuroscience and cell adhesion mechanisms, and her insightful suggestions.

## Additional information

### Funding

| Funder | Grant reference number | Author |
| --- | --- | --- |
| National Institutes of Health | R01GM117916 | Alissa M Weaver |
| National Institutes of Health | R01CA206458 | Alissa M Weaver |
| National Institutes of Health | R01CA249684 | Alissa M Weaver |
| National Institutes of Health | R01CA249424 | Alissa M Weaver |
| American Heart Association | 17POST33660473 | Caitlin McAtee |
| National Institutes of Health | T32CA009592 | Caitlin McAtee |
| Japan Society for the Promotion of Science | JP 17K15005 | Daisuke Hoshino |
| National Institutes of Health | R50CA283661 | Bong Hwan Sung |

The funders had no role in study design, data collection and interpretation, or the decision to submit the work for publication.

## Author contributions
Caitlin McAtee, Conceptualization, Data curation, Formal analysis, Funding acquisition, Validation, Investigation, Visualization, Methodology, Writing – original draft, Writing – review and editing; Mikin R Patel, Conceptualization, Data curation, Validation, Investigation, Visualization, Methodology, Writing – original draft, Writing – review and editing; Daisuke Hoshino, Conceptualization, Data curation, Formal analysis, Validation, Investigation, Visualization, Methodology, Writing – original draft, Writing – review and editing; Bong Hwan Sung, Investigation, Visualization, Methodology, Writing – review and editing; Ariana K von Lersner, Evan Krystofiak, Investigation, Methodology, Writing – review and editing; Mingjian Shi, Investigation; Nan Hyung Hong, Validation, Investigation, Writing – review and editing; Anna Young, Formal analysis; Andries Zijlstra, Resources, Supervision, Methodology, Writing – original draft, Writing – review and editing; Alissa M Weaver, Conceptualization, Resources, Supervision, Funding acquisition, Writing – original draft, Project administration, Writing – review and editing

## Author ORCIDs
Alissa M Weaver ⬤ https://orcid.org/0000-0002-4096-8636

## Ethics
The use of rats for the isolation of embryonic neurons was conducted under the Vanderbilt Institutional Animal Care and Use Committee (IACUC)-approved Protocol #M1800027-00 to ensure humane care and treatment, and all experiments adhered to all relevant federal, state, and local laws and regulations. Experiments were designed to follow the 3Rs principle by minimizing animal use, employing alternative methods where possible, and refining procedures to reduce discomfort. Animals were housed and received appropriate care from the Vanderbilt Division of Animal Care.

Joint Public Review: https://doi.org/10.7554/eLife.101673.3.sa1
Author response https://doi.org/10.7554/eLife.101673.3.sa2

---

# Additional files

## Supplementary files
Supplementary file 1. Proteomics results from bands cut from colloidal blue gel in *Figure 4A*.
Supplementary file 2. iTRAQ proteomics results from control and endoglin-KD B16F1 SEVs.
MDAR checklist

## Data availability
Two proteomics datasets were generated, and both the raw and analyzed data are provided in *Supplementary files 1 and 2*. All data generated or analyzed are included in the figures; source data files have been provided for all gels and blots in the manuscript, including *Figure 2—figure supplement 1*, *Figure 2—figure supplement 2*, *Figure 3—figure supplement 2*, *Figure 4*, *Figure 4—figure supplement 1*, *Figure 4—figure supplement 2*, *Figure 6*, *Figure 6—figure supplement 1*, *Figure 7* and *Figure 8—figure supplement 1*.

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
