## [Editor Report · eLife Assessment]

In this **important** manuscript, the authors reveal novel findings on the role of exosomes in regulating filopodia formation. Filopodia are crucial for various cellular processes, including migration, polarization, directional sensing, and the formation of neuronal synapses. The authors **convincingly** demonstrate that exosomes, particularly those enriched with the protein THSD7A, play a significant role in promoting filopodia formation in both cancer cells and neurons.

---

## [Referee Report · Joint Public Review]

Summary:

The authors identify a novel relationship between exosome secretion and filopodia formation in cancer cells and neurons. They observe that multivesicular endosomes (MVE)-plasma membrane (PM) fusion is associated with filopodia formation in HT1080 cells and that MVEs are present on filopodia in primary neurons. Using overexpression and knockdown (KD) of Rab27/HRS in HT1080 cells, melanoma cells and/or primary rat neurons, they find that decreasing exosome secretion reduces filopodia formation, while Rab27 overexpression leads to the opposite result. Furthermore, the decreased filopodia formation is rescued in the Rab27a/HRS KD melanoma cells by the addition of small extracellular vesicles (EVs) but not large EVs purified from control cells. The authors identify endoglin as a protein unique to small EVs secreted by cancer cells when compared to large EVs. KD of endoglin reduces filopodia formation and this is rescued by the addition of small EVs from control cells and not by small EVs from endoglin KD cells. Based on the role of filopodia in cancer metastasis, the authors then investigate the role of endoglin in cancer cell metastasis using a chick embryo model. They find that injection of endoglin KD HT1080 cells into chick embryos gives rise to less metastasis compared to control cells - a phenotype that is rescued by the co-injection of small EVs from control cells. Using quantitative mass spectrometry analysis, they find that thrombospondin type 1 domain containing 7a protein (THSD7A) is down regulated in small EVs from endoglin KD melanoma cells compared to those from control cells. They also report that THSD7A is more abundant in endoglin KD cell lysate compared to control HT1080 cells and less abundant in small EVs from endoglin KD cells compared to control cells, indicating a trafficking defect. Indeed, using immunofluorescence microscopy, the authors observe THSD7A-mScarlet accumulation in CD63-positive structures in endoglin KD HT1080 cells, compared to control cells. Finally, the authors determine that exosome-secreted THSD7A induces filopodia formation in a Cdc42-dependent mechanism.

Strengths:

Through proteomic analysis, the authors revealed that endoglin is an important player in the effective trafficking of THSD7A within exosomes. This study offers interesting insights into the dynamic interplay between exosome-mediated protein trafficking and essential cellular processes, emphasizing its significant relevance in both cancer progression and neural function. The authors communicated their findings clearly and effectively.

(1) While exosomes are known to play a role in cell migration and autocrine signaling, the relationship between exosome secretion and the formation of filopodia is novel.

(2) The authors identify an exosomal cargo protein, THSD7A, which is essential for regulating this function.

(3) The data presented provide strong evidence of a role for endoglin in the trafficking of THSD7A in exosomes.

(4) The authors associate this process with functional significance in cancer cell metastasis and neurological synapse formation, both of which involve the formation of filopodia.

(5) The data are presented clearly, and their interpretation appropriately explains the context and significance of the findings.

Weaknesses:

While the authors showed the important role of exosomal cargo protein THSD7A in neurons, it will be interesting to conduct any in vivo studies to determine whether THSD7A plays a similar role in promoting filopodia and synapse formation in vivo. Some of the comments of the reviewers were not fully addressed, such as rigorous analysis and quantification through Live-cell imaging through TIRF microscopy tracking labeled THSD7A and filopodia formation, which would provide more clarity in timing and strengthen causality of this relationship. The authors need to consider fully characterizing the role of Cdc42. If the authors would like to fully elaborate on the role of Cdc42 in another manuscript, it is better not to mention at all the role of Cdc42 in filopodia formation in this paper.

---

## [Author Response]

The following is the authors’ response to the original reviews

**Public Reviews:**

**Reviewer #1 (Public review):**
Summary:The study significantly advances our understanding of how exosomes regulate filopodia formation. Filopodia play crucial roles in cell movement, polarization, directional sensing, and neuronal synapse formation. McAtee et al. demonstrated that exosomes, particularly those enriched with the protein THSD7A, play a pivotal role in promoting filopodia formation through Cdc42 in cancer cells and neurons. This discovery unveils a new extracellular mechanism through which cells can control their cytoskeletal dynamics and interaction with their surroundings. The study employs a combination of rescue experiments, live-cell imaging, cell culture, and proteomic analyses to thoroughly investigate the role of exosomes and THSD7A in filopodia formation in cancer cells and neurons. These findings offer valuable insights into fundamental biological processes of cell movement and communication and have potential implications for understanding cancer metastasis and neuronal development.Weaknesses:The conclusions of this study are in most cases supported by data, but some aspects of data analysis need to be better clarified and elaborated. Some conclusions need to be better stated and according to the data observed.

We appreciate the reviewer's recognition of the impact of our study. We will address the concerns about data analysis and the statement of our conclusions in our full response to reviewers.

**Reviewer #2 (Public review):**
Summary:The authors show that small EVs trigger the formation of filopodia in both cancer cells and neurons. They go on to show that two cargo proteins, endoglin, and THSD7A, are important for this process. This possibly occurs by activating the Rho-family GTPase CDC42.Strengths:The EV work is quite strong and convincing. The proteomics work is well executed and carefully analyzed. I was particularly impressed with the chick metastasis assay that added strong evidence of in vivo relevance.Weaknesses:The weakest part of the paper is the Cdc42 work at the end of the paper. It is incomplete and not terribly convincing. This part of the paper needs to be improved significantly

We appreciate the reviewer's recognition of the impact of our study. Indeed, more work needs to be done to clarify the role of Cdc42 in the induction of filopodia by exosome-associated THSD7A. We anticipate that this will be a separate manuscript, delving in-depth into how exosome-associated THSD7A interacts with recipient cells to activate Cdc42 and carrying out a variety of assays for Cdc42 activation.

**Reviewer #3 (Public review):**
Summary:The authors identify a novel relationship between exosome secretion and filopodia formation in cancer cells and neurons. They observe that multivesicular endosomes (MVE)-plasma membrane (PM) fusion is associated with filopodia formation in HT1080 cells and that MVEs are present in filopodia in primary neurons. Using overexpression and knockdown (KD) of Rab27/HRS in HT1080 cells, melanoma cells, and/or primary rat neurons, they found that decreasing exosome secretion reduces filopodia formation, while Rab27 overexpression leads to the opposite result. Furthermore, the decreased filopodia formation is rescued in the Rab27a/HRS KD melanoma cells by the addition of small extracellular vesicles (EVs) but not large EVs purified from control cells. The authors identify endoglin as a protein unique to small EVs secreted by cancer cells when compared to large EVs. KD of endoglin reduces filopodia formation and this is rescued by the addition of small EVs from control cells and not by small EVs from endoglin KD cells. Based on the role of filopodia in cancer metastasis, the authors then investigate the role of endoglin in cancer cell metastasis using a chick embryo model. They find that injection of endoglin KD HT1080 cells into chick embryos gives rise to less metastasis compared to control cells - a phenotype that is rescued by the co-injection of small EVs from control cells. Using quantitative mass spectrometry analysis, they find that thrombospondin type 1 domain containing 7a protein (THSD7A) is downregulated in small EVs from endoglin KD melanoma cells compared to those from control cells. They also report that THSD7A is more abundant in endoglin KD cell lysate compared to control HT1080 cells and less abundant in small EVs from endoglin KD cells compared to control cells, indicating a trafficking defect. Indeed, using immunofluorescence microscopy, the authors observe THSD7A-mScarlet accumulation in CD63-positive structures in endoglin KD HT1080 cells, compared to control cells. Finally, the authors determine that exosome-secreted THSD7A induces filopodia formation in a Cdc42-dependent mechanism.Strengths:(1) While exosomes are known to play a role in cell migration and autocrine signaling, the relationship between exosome secretion and the formation of filopodia is novel.(2) The authors identify an exosomal cargo protein, THSD7A, which is essential for regulating this function.(3) The data presented provide strong evidence of a role for endoglin in the trafficking of THSD7A in exosomes.(4) The authors associate this process with functional significance in cancer cell metastasis and neurological synapse formation, both of which involve the formation of filopodia.(5) The data are presented clearly, and their interpretation appropriately explains the context and significance of the findings.Weaknesses:(1) A better characterization of the nature of the small EV population is missing:It is unclear why the authors chose to proceed to quantitative mass spectrometry with the bands in the Coomassie from size-separated EV samples, as there are other bands present in the small EV lane but not the large EV lane. This is important to clarify because it underlies how they were able to identify THSD7A as a unique regulator of exosome-mediated filopodia formation. Is there a reason why the total sample fractions were not compared? This would provide valuable information on the nature of the small and large EV populations.

We would like to clarify that there are two sets of proteomics data in the manuscript. The first was comparing bands from a colloidal Coomassie-stained gel from two samples: small EVs and large EVs from B16F1 cells. In this proteomics experiment, we identified endoglin as present in small EVs, but not large EVs. For this experiment, we only sent four bands from the small EV lane, chosen based on their obvious banding pattern difference on the Coomassie gel.

In the second proteomics experiment, we used quantitative iTRAQ proteomics to compare small EVs purified from B16F1 control (shScr) and endoglin KD (shEng1 and shEng2) cell lines. In this experiment, we sent total protein extracted from small EV samples for analysis. So, these samples included the entire EV content, not just selected bands from a gel. In this experiment, we identified THSD7A as reduced in the shEng small EVs.

(2) Data analysis and quantification should be performed with increased rigor:a) Figure 1C - The optical and temporal resolution are insufficient to conclusively characterize the association between exosome secretion and filopodia. Specifically, the 10-second interval used in the image acquisitions is too close to the reported 20-second median time between exosome secretion and filopodia formation. Two-5 sec intervals should be used to validate this. It would also be important to correlate the percentage of filopodia events that co-occur with exosome secretion. Is this a phenomenon that occurs with most or only a small number of filopodia? Additionally, resolution with typical confocal microscopy is subpar for these analyses. TIRF microscopy would offer increased resolution to parse out secretion events. As the TIRF objective is listed in the Methods section, figure legends should mention which images were acquired using TIRF microscopy.

We acknowledge that the frame rate naturally limits our estimates of the timing of filopodia formation after exosome secretion. We set out to show a relationship between exosome secretion and filopodia formation, based on their proximity in timing. While our data set shows a median time interval of 20 seconds, the true median could be between 10-30 seconds, based on our frame rate. Regardless of the exact timing, our data show that exosome secretion is rapidly followed by filopodia formation events.

To address the question of the percentage of filopodia events that are preceded by exosome secretion, the reviewer is correct in stating that we might need TIRF microscopy and a faster frame rate to observe all the MVB fusion events and get an accurate calculation of this number. The timing of the acquisition was based on the typical timing of filopodia formation, which is slow relative to MVB fusion. Thus, with the current dataset, we could miss secretion events taking place between the 10 second time intervals. Therefore, to address this question, we would need to acquire a new dataset with a much more rapid frame acquisition (multiple frames per second rather than one frame every ten seconds). Regardless, for the secretion events that we visualized with the current dataset, we always observed subsequent filopodia formation.

No TIRF imaging was used in this manuscript. A TIRF objective was used for selected neuron imaging (see methods); however, it was used for spinning disk confocal microscopy, not for TIRF imaging. This is stated in the methods.

b) Figure 2 - It would be important to perform further analysis to concretely determine the relationship between exosome secretion and filopodia stability. Are secretion events correlated with the stability of filopodia? Is there a positive feedback loop that causes further filopodia stability and length with increased secretion? Furthermore, is there an association between the proximity of secretion with stability? Quantification of filopodia more objectively (# of filopodia/cell) would be helpful.

Our data show that manipulation of general exosome secretion, via Hrs knockdown, affects both de novo filopodia formation and filopodia stability (Fig 2g,h). Interestingly, knockdown of endoglin only affects de novo filopodia formation, while filopodia stability is unaffected (Fig 4g,h). These results suggest that filopodia stability is dependent upon exosome cargoes besides endoglin/THSD7A. Such cargoes might include other extracellular matrix molecules, such as fibronectin. We previously showed that exosomes promote nascent cell adhesion and rapid cell migration, through exosome-bound fibronectin (Sung et al., Nature Communications, 6:7164, 2015). We also previously found that inhibition of exosome secretion affects the persistence of invadopodia, which are filopodia-dependent structures (Hoshino et al., Cell Reports, 5:1159-1168, 2013). We agree that this is an interesting research direction, and perhaps future work could focus on exosomal factors that are responsible for filopodia persistence. This would possibly involve more proteomics analysis to identify candidate exosomal cargoes involved in this process.

With regard to the way we plotted the filopodia data, we plotted the cancer cell data as filopodia per cell area so that it matched the neuron data, which was plotted as filopodia per 100 µm of dendrite distance. Since the neurons cannot be imaged as a whole cell, the quantification is based on the length of the dendrite in the image. We found that graphing the cancer cell data as filopodia per cell gave similar results as filopodia per cell area. To demonstrate that this quantification gives similar results, we have now plotted the filopodia per cell area data from Fig 2 as filopodia per cell and placed these new plots in Supp Fig 2.

c) Figure 6 - Why use different gel conditions to detect THSD7A in small EVs from B16F1 cells vs HT1080 and neurons? Why are there two bands for THSD7A in panels C and E? It is difficult to appreciate the KD efficiency in E. The absence of a signal for THSD7A in the HT1080 shEng small EVs that show a signal for endoglin is surprising. The authors should provide rigorous quantification of the westerns from several independent experimental repeats.

Detection of THSD7A via Western blot was, unfortunately, not straightforward and simple. Due to the large size (~260 kDa) of THSD7A, its low level of expression in cancer cells, as well as the inconsistency of commercially available THSD7A antibodies, we had to troubleshoot multiple conditions. We found that it was much easier to detect THSD7A in the human fibrosarcoma cell line HT1080 than in the mouse B16F1 cells, both in the cell lysates and in the small EVs. We were unable to detect THSD7A using the same (reducing) conditions for the mouse melanoma B16F1 samples but were successful using native gel conditions. We also detected THSD7A in rat primary neuron samples. All these samples were from different source organisms (human, mouse, rat) and from either cell lysates or extracellular vesicles, further complicating the analyses. Expression and maturation of THSD7A in these different cell types and compartments could involve different post-translational modifications, such as glycosylation, thus requiring different methods needed to detect THSD7A on Western blots and leading to different banding patterns.

With regard to the level of knockdown of THSD7A in the Western blot shown in Figure 6E, the normalized level is quantitated below the bands. If you compare that quantitation to the filopodia phenotypes in the same panel, they are quite concordant. Figures 7B and 7C show quantification of triplicate Western blots, highlighting the significant accumulation of THSD7A in shEng cell lysates, as well as significant small EV secretion of THSD7A in control and WT rescued conditions.

(3) The study lacks data on the cellular distribution of endoglin and THSD7A:a) Figure 6 - Is THSD7A expected to be present in the nucleus as shown in panel D (label D is missing in the Figure). It is not clear if this is observed in neurons. a Western of endogenous THSD7A on cell fractions would clarify this. The authors should further characterize the cellular distribution of THSD7A in both cell types. Similarly, the cellular distribution of endoglin in the cancer cells should be provided. This would help validate the proposed model in Figure 8.

The image in figure 6D shows an HT1080 cell stained with phalloidin-Alexa Fluor 488 to visualize F-actin with or without expression of THSD7A-mScarlet. In order to fully visualize the thin filopodia protrusions, the cellular plane of focus of the images for this panel was purposely taken at the bottom of the cell, where the cell is attached to the coverslip glass. Thus, we interpret the red signal across the cell body as THSD7A-mScarlet expression on the plasma membrane underneath the cell, not in the nucleus. The neuron images only include the dendrite portion of the neurons; therefore, there is no nucleus present in the neuronal images. For the cellular distribution of endoglin, we agree that this is an important future direction to understand how endoglin regulates THSD7A trafficking. We have added the lack of these data to the “Limitations” section at the end of the manuscript.

b) Figure 7 - Although the western blot provides convincing evidence for the role of endoglin in THSD7A trafficking, the microscopy data lack resolution as well as key analyses. While differences between shSCR and shEng cells are clear visually, the insets appear to be zoomed digitally which decreases resolution and interferes with interpretation. It would be crucial to show the colocalization of endoglin and THSD7A within CD63-postive MVE structures. What are the structures in Figure 7E shSCR zoom1? It would be important to rule out that these are migrasomes using TSPAN4 staining. More information on how the analysis was conducted is needed (i.e. how extracellular areas were chosen and whether the images are representative of the larger population). A widefield image of shSCR and shEng cells and DAPI or HOECHST staining in the higher magnification images should be provided. Additionally, the authors should quantify the colocalization of external CD63 and mScarlet signals from many independently acquired images (as they did for the internal signals in panel F). Is there no external THSD7A signal in the shEng cells?

The images for Figure 7E were taken with high resolution on a confocal microscope. Insets for Figure 7E were digitally zoomed so that readers could see the tiny structures. Zoom 1 in Figure 7E shows areas of extracellular deposition, whereas Zoom 2 shows THSD7A colocalization with CD63 in MVE. In the extracellular areas (Zoom 1), we observe small punctate depositions that are positive for CD63 and/or THSD7A-mScarlet. Our interpretation of this staining is that the cells are secreting heterogeneous small EVs that are then attached to the glass coverslip. The images and zooms in Fig 7E were chosen to be representative and indeed reveal that there is more extracellular deposition of THSD7A-mScarlet outside the control shScr cells compared to the shEng cells, consistent with more secretion of THSD7A in small EVs from shScr cells when compared to those of shEng cells (Fig 7A,B). However, we did not quantify this difference, as these experiments were conducted with transient transfection of THSD7A-mScarlet, and it is challenging to determine which cell the extracellular THSD7A-mScarlet came from, complicating any quantitative analysis on a per-cell basis.

Quantification of internal THSD7A localization is much more straightforward in this experimental regime. Indeed, in Figure 7F, we quantitated internal colocalization of THSD7A-mScarlet and CD63, which we obtained by choosing only cells that were visually positive for THSD7A-mScarlet in each transient transfection and omitting all extracellular signals. Quantifying the extracellular colocalization of THSD7A and CD63 could certainly be a future direction for this project and would require establishing cells that stably express THSD7A-mScarlet.

With regard to whether the extracellular deposits are migrasomes, we have no reason to believe that they would be migrasomes. The preponderance of our evidence points to exosomes as carrying THSD7A and inducing filopodia. Furthermore, CD63 is an exosome marker (Sung et al., Nat Comm, 2020) and does not induce migrasomes, unlike many other tetraspanins (Huang et al., Nat Cell Bio, 2019).

**Recommendations for the authors:**

**Reviewer #1 (Recommendations for the authors):**
The authors need to clarify the figure labeling and description and conclusions would be better to be drawn based on the findings. Some figures need to more clear e.g. Figure 1E needs to have information on what are green and red fluorescent proteins. Do all figures in 1A have the same scale bar or different? Figure 3A lacks a scale bar. In Figure 3, the GFP signal is in yellow, does it represent a merge or is it just the GFP alone? Figure 6D is missing a D. Figure 4D needs to be better explained. Additionally, both figures 8B and 8C since represent a model based on all the findings of the study would be better to stand alone as a separate figure from figure 8A.

The figure legend for figure 1E notes that green corresponds to GFP-Rab27b and the red corresponds to mCherry filler. In addition, the labels are marked to the right of the figure. For Figure 1A, we have now indicated in the legend that all scale bars = 10 µm. In figure 3, neurons were co-transfected with GFP or GFP-Rab27b. Thus, the yellow signal in these images is the merge of the mCherry filler with either GFP (expression throughout the neuron body and dendrites) or GFP-Rab27b (punctate colocalization). We have added a scale bar to Fig 3A. Figure 6D has been corrected, with a “D” label added. Figure 4D shows representative images of cells with filopodia under the various conditions, including add-back of control or endoglin-KD EVs. We have clarified the conditions in the figure legend for 4D. For Figure 8, we have now split it into 2 figures: one with data (Fig 8) and one with the model (Fig 9).

**Reviewer #2 (Recommendations for the authors):**
For the most part, this story is strong and well-presented. The findings are interesting and will significantly advance our understanding of how EVs affect various processes such as cancer metastasis. However, the Cdc42 work is not great. They only indirectly implicate Cdc42 with a somewhat iffy inhibitor (ML141) and a constitutively active form transfected into cells. Both approaches have drawbacks such as off-target effects in the case of the inhibitor and possible cross-talk to other GTPases in the case of the active mutant. The activation of Cdc42 should be demonstrated by an activity assay. Several commercial kits are available. Inhibition of Cdc42 should be tested by knockdown in addition to the inhibitor.

We appreciate the reviewer’s recognition of our work. To address the limitations of our study, particularly the Cdc42 mechanistic work, we have now added a “Limitations of the study” section at the end of the text. Here, we address our experimental limitations and future directions.

**Reviewer #3 (Recommendations for the authors):**
(1) Since the purified small EVs contain canonical exosomal markers and originate from MVEs, the authors should consider a more consistent use of the term "exosome" to avoid confusion.

We acknowledge that the usage of both “exosomes” and “small extracellular vesicles” can seem confusing to many readers. Typically in the EV field, we use the term “exosome” when we can reliably determine that the EVs originate from the endocytic pathway. Thus, we use this term when we have specifically perturbed this pathway by targeting Hrs or Rab27. We use the term “small extracellular vesicles” or SEVs when referring to a purified heterogeneous population of SEVs from unknown or a variety of origins. Thus, when referring to vesicles isolated from the conditioned media, we call them SEVs because we cannot determine their origin. Clarification of this terminology has been added to the introduction of the paper.

(2) 1st results section - expressing mCherry as a "filler" is confusing, clarify that this is meant to identify cellular background.

This has now been clarified in the paper.

(3) Figure 3 - Although Rab27a and Rab27b play a role in exosome secretion, Rab27b does not have redundant functions with Rab27a in every cellular context. The authors should mention the specific roles of Rab27a and Rab27b in promoting MVE fusion with the PM and in regulating the anterograde movement of MVEs to the PM, respectively (Ostrowski et al. 2010, Citation 52 in the ms). Although Rab27a is not highly expressed in neurons, it is not currently clear whether Rab27b has a redundant function with Rab27a or whether there is another unknown factor that plays this role. As neurons also do not express endoglin, the mechanisms that mediate how EVs regulate filopodia formation in these cells are most probably different than in cancer cells. This should be highlighted in the discussion.

We have now added a couple of clarifying sentences about the roles of Rab27a and Rab27b to the results section, including the Ostrowski reference and another reference suggesting possible redundancy of Rab27a and Rab27b. With regard to endoglin not being expressed by neurons, that is one reason why we carried out the proteomics with control and endoglin-KD EVs to find a universal cargo that would directly induce filopodia formation. Indeed, THSD7A seems to be such a universal cargo, expressed in both cancer cell and neuron EVs and inducing filopodia in both cell types. This point, along with the requirement for regulation of THSD7A by other molecules in neurons, is discussed in the results and discussion sections.

(4) As the authors note, the mechanistic link between endoglin-sorted, exosomal THSD7A and Cdc42-mediated filopodia formation remains unclear. While the findings on Cdc-42 are clear, they are not surprising. What is the role of mDia/ENA/VASP or BAR proteins in this? The authors should also consider an assay to determine whether exosomal THSD7A binds to the PM to cause the signaling or if the cargo is first internalized before performing its function. Since this process is both autocrine and paracrine, the authors could co-culture THSD7A-mScarlet cells with vector control cells and observe how THSD7A-mScarlet is localized in the non-expressing cells.

As other reviewers also noted, the Cdc42 mechanistic data at the end of the paper has clear limitations that are now addressed within the manuscript in a “Limitations of the Study” section. Here we discuss our experimental troubleshooting and approach to assaying Cdc42 involvement in this process. We acknowledge there are many rigorous experiments that could be pursued in the future to strengthen our mechanism and proposed model.

We also agree that elucidating how THSD7A specifically interacts with target cells would be very informative and insightful. This would be most effectively assayed using a cell line that is stably expressing THSD7A-mScarlet and could be a future direction of this project. However, it is out of the scope of this current publication.